# Sediment size on talus slopes correlates with fracture spacing on bedrock cliffs: Implications for predicting initial sediment size distributions on hillslopes

Joseph P. Verdian[1], Leonard S. Sklar[1,2], Clifford S. Riebe[3], Jeffrey R. Moore[4]

[1]Department of Earth and Climate Sciences, San Francisco State University, San Francisco, 94302, USA
[2]Department of Geography, Planning and Environment, Concordia University, Montreal, H3G1M8, Canada
[3]Department of Geology and Geophysics, University of Wyoming, Laramie, 82071, USA
[3]Department of Geology and Geophysics, University of Utah, Salt Lake City, 84112, USA

*Correspondence to*: Leonard S. Sklar (leonard.sklar@concordia.ca)

**Abstract.** The detachment of rock fragments from fractured bedrock on hillslopes creates sediment with an initial size distribution that sets the upper limits on particle size for all subsequent stages in the evolution of sediment in landscapes. We hypothesize that the initial size distribution should depend on the size distribution of latent sediment (i.e., fracture-bound blocks in unweathered bedrock) and weathering of blocks both before and during detachment (e.g., disintegration along crystal grain boundaries). However, the initial size distribution is difficult to measure, because the interface across which sediment is produced is often shielded from view by overlying soil. Here we overcome this limitation by comparing fracture spacings measured from exposed bedrock on cliff faces with particle size distributions in adjacent talus deposits at 15 talus-cliff pairs spanning a wide range of climates and lithologies in California. Median fracture spacing and particle size vary by more than tenfold and correlate strongly with lithology. Fracture spacing and talus size distributions are also closely correlated in central tendency, spread, and shape, with b-axis diameters showing the closest correspondence with fracture spacing at most sites. This suggests that weathering has not modified latent sediment either before or during detachment from the cliff face. In addition, talus at our sites has not undergone much weathering after deposition and is slightly coarser than the latent sizes, because it contains unexploited fractures inherited from bedrock. We introduce a new conceptual framework for understanding the relative importance of latent size and weathering in setting initial sediment size distributions in mountain landscapes. In this framework, hillslopes exist on a spectrum defined by the ratio of two characteristic timescales: the residence time in saprolite and weathered bedrock, and the time required to detach a particle of a characteristic size. At one end of the spectrum, where weathering residence times are negligible, the latent size distribution can be used to predict the initial size distribution. At the other end of the spectrum, where weathering residence times are long, the latent size distribution can be erased by weathering in the critical zone.

## 1 Introduction

The detachment of rock fragments from fractured or weathered bedrock creates sediment particles of various sizes that evolve due to chemical and physical weathering during transport and storage on slopes and in rivers. At all stages in the evolution of sediment, its size distribution influences local chemical, physical, and biological processes, including throughflow of reactive fluids in soil (Maher, 2010; Brantley et al., 2011), river incision into bedrock (Sklar and Dietrich, 2004; Turowski et al., 2015; Shobe et al., 2016), and the reproductive potential of aquatic habitat (Riebe et al., 2014; Overstreet et al., 2016). The first stage begins when particles are initially detached from saprolite (in soil-mantled landscapes) or fractured bedrock (when soil is absent). At any specific source location, whether it's at the surface or at the base of soil, a set of these newly detached sediment particles have an "initial size distribution," where initial is defined by the moment when particles are first disturbed by processes that drive particle motion on hillslopes (Sklar et al., 2017; Roda-Boluda et al., 2018; Neely and DiBiase, 2020). The initial size distribution sets the initial condition for subsequent evolution of the size distribution by abrasion and fragmentation of particles during transport, both on hillslopes near where sediment is produced and in river channels downstream. The initial size distribution is therefore a fundamental control on sediment size distributions across landscapes. It sets the upper limit for sediment size in a catchment; in the absence of flocculation or cementation, sediment particles can only become smaller. In addition, the initial size sets the scale for particle size reduction by comminution during transport away from its source. The downstream evolution of sediment size is commonly modeled as a function of three factors: the initial size, the travel distance downstream from the source, and a rate constant that depends primarily on rock durability (Sklar et al., 2006; Attal and Lavé, 2009; Dingle et al., 2017). Thus, a coarser initial size will result in a coarser downstream size, for a given transport distance and rock durability. Particle shape also evolves during transport downstream and can be used to estimate the distance traveled from the particle's source when initial shape is known (Miller et al., 2014; Szabo et al., 2015). However, the factors that regulate variability in the initial size and shape of sediment are poorly understood.

The initial size distribution and initial particle shape should depend on three factors: the size distribution of discrete blocks latent within bedrock; the weathering processes that modify rock in situ; and the weathering processes that detach and mobilize the sediment. Where effects of weathering on size are minimal, the initial size distribution should strongly reflect the size distribution of latent blocks (i.e. the "latent size distribution"), which is set by the spacing and orientation of fractures, foliations, and bedding planes (Sklar et al., 2017). These represent preexisting planes of weakness and determine the volume and shape of newly created sediment particles, which can be quantified by measuring the distributions of the major-, intermediate-, and minor-axis particle diameters. The three-dimensional template for latent particles should depend on conditions experienced during formation of the rock at depth, including rate of cooling for igneous rocks (Lore et al., 2001), pressure and temperature for metamorphic rocks (Manda et al., 2008), and deposition and diagenesis in sedimentary rocks (Narr and Suppe, 1991). These factors are overprinted by fracturing induced by the evolving stress field experienced by the rock as it is exhumed from deep in the crust (Molnar et al., 2007; Leith et al., 2014; Moon et al., 2017). Thus, the latent size distribution reflects everything that has happened to the rock before it arrives at the base of the critical zone, where it is first

exposed to near-surface conditions that promote weathering (Riebe et al., 2017). Where near-surface residence times are long or weathering is intense, the initial size distribution can be affected by in-situ weathering during exhumation through the critical zone (Sklar et al., 2017). For example, chemical reactions such as mineral dissolution and physical processes such as mineral expansion can create new planes of weakness and thus influence the initial size distribution created during detachment (Fletcher and Brantley, 2010; Brantley et al., 2011; Shen et al., 2019). In-situ chemical and physical weathering can also modify rock strength, density, and other properties that might affect initial size without altering the apparent fracture spacing (Goodfellow et al., 2016; Riebe et al., 2021). In addition to these in-situ weathering processes, the initial size distribution should also depend on the characteristic length scales of weathering processes that detach and mobilize sediment, such as segregation ice growth, root wedging, and animal burrowing (Sklar et al., 2017; Messenzehl et al., 2018), particularly on slopes where preexisting planes of weakness in bedrock are widely spaced (Marshall and Roering, 2014). Although these hypotheses are both intuitive and mechanistic, to our knowledge, the relative importance of latent sizes, in-situ weathering, and detachment processes in initial size distributions has not been systematically explored. The initial detachment of rock fragments is generally hidden from view under soil, and particle sizes can therefore evolve during mixing, transport, and storage, before they are measured. This makes it difficult to isolate the influence of the latent size distribution from the effects of weathering processes.

To overcome this limitation, previous studies have compared fracture spacings measured from exposed bedrock on cliff faces with particle size distributions in adjacent sediment deposits. For example, at Inyo Creek, on the east side of the Sierra Nevada, California, where hillslope sediment size distributions are bimodal, in-situ measurements from bedrock cliff faces show that fracture spacing distributions do not vary with elevation and closely correspond with the likewise spatially invariant coarse mode of the hillslope particle size distribution (Sklar et al., 2020). Similarly, in California's San Gabriel and San Jacinto mountains, two regions with differing fracture spacing distributions, particle sizes in stream sediment correlate with fracture spacings measured in adjacent bedrock cliff faces at locations where the sediment contributing area is dominated by bare bedrock (DiBiase et al., 2018; Neely et al., 2019; Neely and DiBiase, 2020). Results from both of these studies are consistent with latent sizes in bedrock dominating over weathering on slopes in regulating the initial size distribution of coarse sediment. In contrast, results from paired cliffs and talus slopes in the Swiss Alps suggest that weathering by frost cracking can impose a characteristic scale upon talus particle sizes (Messenzehl et al., 2018). At these sites, talus size is roughly uniform across a suite of locations where fracture spacing differs by over an order of magnitude. Measured talus size is always either larger than or approximately equal to the mean fracture spacing, consistent with the hypothesis that talus production by frost-cracking can preferentially exploit a subset of pre-existing fractures with a characteristic spacing, leaving other fractures intact within the blocks detached from the cliff wall. Variation in the frequency and intensity of frost cracking with elevation at the Inyo Creek site may also be implicated in the downvalley fining of the fine mode of the hillslope sediment size distribution (Riebe et al., 2015; Sklar et al., 2020). The results from California and the Swiss Alps show that both the latent size distribution and sediment detachment mechanisms can strongly influence initial sediment size, when in-situ weathering is minimized. Quantifying the relative importance of these factors at additional sites is vital to predicting initial sediment sizes across a range of conditions.

Here we quantify correlations between initial sediment size and fracture spacing distributions across 15 talus-cliff pairs spanning a wide range of settings, including rock types and climatic conditions not investigated in previous work. Our sites span a 3-km range in elevation across granodiorite, andesite, basalt, metasedimentary, and chert lithologies in California. All cliffs are sufficiently steep to prevent any accumulation of regolith, such that surface fracture spacing can be assumed to reflect the latent size distribution with minimal influence of weathering. Thus our study design allows us to test the null hypothesis that the initial sizes closely match latent sizes across a wide, geologically-driven range in latent size distributions. An alternate hypothesis is that talus size distributions and the latent size distributions exposed in cliff faces are not strongly correlated for one or more of the following reasons: (i) blocks are detached along only a subset of preexisting fractures, because of characteristic scales of detachment processes such as frost cracking (Messenzehl et al., 2018), or due to unequal fracture persistence (Kim et al., 2007); (ii) talus particles are created by detachment along newly-formed fractures, as in the case of grus production from granite with low fracture density (Wahrhaftig, 1965); (iii) physical or chemical weathering reduces particle sizes after the talus is detached from the cliff, for example when falling particles impact the slope below or as they sit in the talus deposit. None of these alternatives to the null hypothesis are consistent with our results. Measurements of central tendency, spread, and shape of the talus size and fracture spacing distributions all correlate strongly across a 40-fold variation in median fracture spacing. We also found statistically significant differences in mean talus shape among rock types, contrary to the null hypothesis that initial particle shape is invariant for blocks produced from bedrock by mechanical weathering (Domokos et al., 2015). Together these results confirm that initial sediment size distributions can be predicted from fracture spacing distributions at sites where the latent size distribution dominates over weathering. They also imply that lithologic and tectonic controls on latent size distributions can have a strong influence on the initial size and shape of individual particles and thus on the evolution of particle size distributions across landscapes. To generalize our findings beyond the talus-cliff pairs studied here, we introduce a conceptual framework for quantifying the relative importance of latent sizes and weathering using the timescale of detachment of latent particles and the timescale of weathering that occurs before the particle is detached.

## 2 Methods

### 2.1 Site selection

To test the null hypothesis that latent size, set by fracture spacing, should control the initial size of sediment produced on bare bedrock slopes, we selected 15 cliff faces and adjacent talus slopes at five sites in California, USA (Fig. 1). The talus-cliff pairs in the Sierra Nevada are a subset of the bedrock slopes used by Moore et al. (2009) to quantify the influence of rock-mass strength on cliff retreat rates. These include ten slopes in the vicinity of Conness Basin, Mount Tallac, and Ebbetts Pass (Table 1), three sites where differences in lithology correspond to differences in average fracture spacing. To diversify the range of conditions that might contribute to differences in weathering and thereby produce deviations from the null hypothesis,

we selected five additional talus-cliff pairs at Grizzly Peak and Twin Peaks, two sites in the San Francisco Bay Area (Table
30    1).


The Sierra Nevada sites include exposures of the three major classes of rock comprising the high elevation terrain of the range
(Bateman et al., 1966). The youngest rocks are volcanic basalts and andesites, exposed at Mount Tallac and Ebbetts Pass,
which erupted during the Miocene Epoch (Armin et al., 1984). These volcanic rocks overlie intrusive granitic rocks
characteristic of the Sierra Nevada batholith, emplaced during the Cretaceous Period, including the granodiorites exposed at
Conness Basin (Bateman, 1983) and Mount Tallac (Saucedo, 2005). Also exposed at Mount Tallac are the oldest rocks of our
study sites, Paleozoic-age metasedimentary roof pendants of the country rock intruded by the Cretaceous granitics (Saucedo,
2005). Bedrock at the San Francisco Bay Area sites is characteristic of the Franciscan Formation, and includes Mesozoic
pillow basalt and overlying chert exposed at Twin Peaks, as well as Miocene subaerial basalt flows exposed at Grizzly Peak
(Graymer et al., 2006).

This spectrum of rock types and geologic settings provides a potentially large range of latent size distributions, from the
widely-spaced fractures in the massive granodiorite at Conness Basin to the closely-spaced bedding planes of the Twin Peaks
ribbon chert. Similarly, fractures in bedrock at our sites are produced by a range of mechanisms, from cooling of subaerial
lava at Grizzly Peak, Ebbetts Pass, and Mount Tallac to the crystallization and exhumation of granodiorite through ~10 km of
crust at Conness Basin (Ague, 1997). In addition, the sites span a range in weathering conditions, due to the 3-km range in
elevation, the 10°C range in mean annual temperature, and the twofold variation in mean annual precipitation (Table 1).
Collectively, this combination of rock types, fracturing mechanisms, and weathering conditions can be expected to produce
talus spanning a wide range of sizes.
**2.2 Fracture spacing distributions on cliff faces**
To quantify fracture spacing at each site, we used a horizontal scan line (Priest and Hudson, 1976; Moore et al., 2009;
Messenzehl et al., 2018) consisting of a survey tape stretched across the cliff face at a constant elevation (Fig. 2a). The height
of the scan line above the top of the adjacent talus cone ranged from 0.3 to 1.5 meters in our study, depending mostly on ease
of sampling. Our approach assumes that fracture spacing along a single horizontal line is representative of the contributing
area of the talus, including unreachable sections above the scan line. We set the length of each scan line equal to the width of
contact between the cliff face and its adjacent talus cone, which ranged between 5- and 15-meters long across our sites. Thus
we limited our measurements of fracture spacing to the width of the talus source area. Along each scan line, we measured the
position of every fracture that crossed the tape, irrespective of orientation. This yields a distribution of fracture spacings
measured as the distance between successive fractures (Fig. 2b). Our goal was to sample the spacing between fractures that
could produce a particle via rockfall. Therefore, we quantified the distance between fractures that were unambiguous partings
in the rock (with aperture generally larger than 0.5 mm) that were also through-going (i.e., persistent enough to intersect other
fractures on the cliff face). We ignored fractures with spacings less than 2 mm and variations in surface roughness and other
rock defects. This approach does not account for the potential role of micro-fracturing at the scale of mineral grains (Eppes
and Keanini, 2017; Eppes et al., 2018) in generating detachable particles on the cliff faces.

## 2.3 Particle size distributions in talus cones

To quantify surface particle size distributions in talus at each of the 10 Sierra Nevada talus-cliff pairs, we measured the a-, b-,
and c-axis diameters of particles at evenly-spaced points along three slope-parallel transects extending from the base of the
cliff to the toe of the talus slope (e.g. Fig. 2). At each talus-cliff pair, we measured a total of 100 particles. To obtain an equal
spacing (and thus obtain a random sample), we divided the sum of the three transect lengths by 100 particles, yielding spacings
that varied from 1.0 to 2.5 m/particle across the sites, with larger spacings at talus-cliff pairs that have longer talus cones. At
each sampling point, we used a ruler to measure sizes of particles with diameters less than 300 mm and stadia rods to measure
sizes for larger particles. In some cases, the a-, b-, and/or c-axis could not be readily measured because the particle was too
heavy to move and thus to fully expose it for identification of long-, intermediate-, and short-axis orientations. In those cases,
we assumed that the c axis was perpendicular to the surface slope and estimated the a- and b-axis diameters using the two
longest exposed axes. We then estimated a minimum value for the c axis as the height of the particle normal to the slope.
The even spacing in our talus sampling approach should yield a representative particle size distribution, even if size-selective
transport leads to downslope coarsening, which is commonly observed on angle-of-repose slopes (Kirkby and Stratham, 1975).
This coarsening arises because finer particles encounter relatively larger friction angles and therefore travel shorter distances
before coming to rest on the talus cone. Because size-selective disentrainment occurs across the entire slope, the talus surface
can be treated as a single population whose grain size distribution can be quantified representatively by uniformly spaced
sampling (Bunte and Abt, 2001).
We addressed the potential for bias due to kinetic sieving (a vertical sorting process) by supplementing the surface-based
measurements with bulk samples of relatively fine subsurface sediment accessed through openings between particles at the
surface at five of the Sierra Nevada talus-cliff pairs. The particle size distribution of each ~2 kg sample was measured in the
lab by standard mechanical sieving. In the absence of substantial vertical sorting, these subsurface size distributions should
overlap with and continuously extend the fine tail of the size distribution from the surface, rather than represent a distinct
population with its own mode (Bunte and Abt, 2001).
At the three San Francisco Bay Area sites, where the talus cones are relatively small, we did not sample along linear transects.
Instead, we measured surface particle size distributions using standard random point counting methods (Bunte and Abt, 2001)
to sample 100 particles from each talus cone. At the Grizzly Peak site, we used rulers to quantify a-, b-, and c-axis diameters
of sampled particles. At the Twin Peaks site, where talus produced from both pillow basalt and ribbon chert were small
compared to talus produced at the other sites, we used a mix of calipers and rulers to quantify just the b-axis diameters.

## 3 Results and Interpretations

Spacings between individual fractures on cliff faces range from 2 to 5000 mm across the suite of sites, with median spacings
at individual sites ranging from 10 to 390 mm (Table 2). Particle sizes span a similar range, with a-axis diameters as large as
5450 mm, c-axis diameters as small as 2 mm, and median b-axis diameters ranging from 10 to 575 mm. Both fracture spacing
and particle size vary systematically with lithology: Granodiorite sites have the largest fracture spacings and particle sizes
while the pillow basalt site has the smallest (Fig. 3).

Across all 15 sites, the distributions of particle sizes in talus generally correspond to the distribution of fracture spacings on
adjacent cliffs. This is evident in both the similar shape and overlap of the size and spacing distributions (Fig. 3). For example,
at most of the sites, the empirical cumulative distribution function (ECDF) of fracture spacing is parallel (i.e. offset by a
constant distance) to the ECDFs of particle size, which are also generally parallel to each other where the a-, b-, and c-axis
diameters were measured together (Fig. 3). In many cases, the ECDFs of size and spacing also overlap for at least one of the
particle diameters. For example, at CB-1, the fracture spacing distribution closely overlaps with the size distribution of the a-
axis particle diameters (Fig. 3e). In contrast, the overlap is closest with the b-axis diameters at EP-26, TP-1, and TP-3, and
with the c-axis diameters at both MT-38 and CB-5. Only two of the sites (CB-2 and CB-3) have particle diameter ECDFs that
do not closely parallel the fracture spacing ECDF, and only one (MT-39) has a fracture spacing ECDF that plots outside the
envelope defined by the a- and c-axis diameters.
The close correspondence between distributions at each talus-cliff pair is reflected in cross-site correlations between the central
tendency of fracture spacing and particle size distributions for each of the three particle axes (Fig. 4). To test for a systematic
correspondence between fracture spacing and talus size we used linear regression to fit trends to the log-transformed medians
of the fracture spacing and particle diameter distributions measured at each site. We then compared the slopes of the best-fit
trend lines to a slope of 1.0, which corresponds to the null hypothesis that there is a 1:1 relationship between median talus size
and median fracture spacing. We found that for each of the three particle axes, the increase in median particle diameter with
increasing fracture spacing (Fig. 4a-c) follows a trend with a slope that is statistically indistinguishable from 1.0 (two-tailed t-
test, $p>0.45$). This result suggests that the correspondence between talus size and fracture spacing is independent of both scale
and rock type across the full range of measured sizes (up to two orders of magnitude for the b-axis diameters).
Although the site-to-site variation in talus size scales with the site-to-site variation in fracture spacing, each of the three talus-
diameter trend lines is offset vertically from an exact 1:1 relationship. Of the three axes, the a-axis diameters have the largest
offsets, with median diameters that are systematically greater than the median fracture spacing by a factor of ~2.5 on average
(Fig. 4a). The c-axis diameters have a smaller offset (52%) and plot below the 1:1 line (Fig. 4c). The offset is smaller still and
positive for the b-axis diameters (Fig. 4b), which are 42% larger on average than the fracture spacing. This offset is equivalent
to a $\Phi/2$ interval, using the common log transformation $\Phi = \log_2(size)$. All but 2 of the 15 site-specific b-axis medians plot
above the 1:1 line, suggesting that the vertical offset is not due to random variation. A binomial test indicates that 2 or fewer
negative offsets would arise by chance just 0.64% of the time when negative and positive offsets are equally likely. This
confirms the systematically higher b axis diameters relative to fracture spacings is likely due to a systematic measurement bias
or a natural process such as preferential detachment along a subset of the measured fractures.

The link between talus and fracture-bound blocks exposed on adjacent cliff faces is further supported by the close site-to-site
correspondence between the spread in the b-axis diameter distribution and the spread in the fracture spacing distribution (Fig.
4d). Because our measurements of fracture spacing and talus b-axis diameter vary over several orders of  magnitude, we used
the geometric standard deviation, a non-dimensional metric, to quantify and compare the spreads in the distributions of both
variables. To test the null hypothesis that site-to-site variation in the spread in b-axis distribution scales directly with the site-
to-site variation in fracture spacing distribution, we compared the slope of a linear trend through the data with a 1:1 slope. We
found that the relationship between the geometric standard deviation of the b-axis diameters and fracture spacing across all
sites is statistically indistinguishable from a 1:1 relationship (two-tailed t-test, p>0.67), a quantitative reflection of  the fact
that the ECDFs of the b-axis diameters and fracture spacings are parallel at many of the sites in Fig. 3. Thus, both the central
tendency and spread in the size and spacing distributions are closely coupled across the range of rock types and climates
spanned by our study sites.

A third aspect of the particle size and fracture spacing distributions that we explored is distribution shape, which can be
quantified using probability density functions. To determine if the fracture spacing and b-axis size distributions share similar
shapes, we first fit exponential, log-normal, and power distribution functions to the data, recognizing that fracture spacing
distributions in rock commonly have shapes that follow one of these distributions (Gillespie et al., 1993). For both the fracture
spacing and talus size distributions, we found that the Weibull form of the exponential distribution (Weibull, 1951) yielded
the best fit to the data in most cases. The degree to which the data follow a Weibull distribution can be evaluated graphically
using a standard linearization of the Wiebull equation (e.g. Nakamura et al., 2007; Litwin et al., 2012), as illustrated in Figure
5. Data sampled from a population that follows a Weibull distribution will plot on a straight line in the Weibull probability
space, which is defined by the plot axes of log-transformed cumulative probability (P) and log-transformed size. In addition,
because we normalized the particle size and fracture spacing measurements by their respective medians, cumulative
distributions that coincide in the plotting space (i.e. have the same slopes and heights) are indicative of population distributions
that have the same shape.

For most sites, both the particle size and fracture spacing data fall along straight lines and often closely coincide, as in the case of MT-38, CB-1, CB-5, and TP-1 (Fig. 5a, 5e, 5f and 5o), indicating that they share roughly the same Weibull distributions. In some cases, the slope of the b-axis distribution is steeper than the fracture spacing distribution, as in the case of MT-39 and CB-3 (Fig. 5b, 5c), indicative of narrower particle size distributions and consistent with the systematically lower geometric standard deviations at these sites (Fig. 4d). In some cases, the lower tails of the distributions follow a steeper trend than the rest of the data, as in the case of CB-3, GP-1, and EP-26 (Fig. 5c, 5j, and 5k), potentially reflecting an undersampling of the smallest fractures that could result from the limited sample size and our emphasis on quantifying spacings of throughgoing fractures in the scan lines. The one rock type with data that deviate substantially from the Weibull distribution is the chert: at TP-3 and especially at TP-4 (Fig. 5l, 5m), the data show curvature in the Weibull space, and the particle size and fracture spacing distributions do not closely match. Aside from these exceptions, the fracture spacing and talus size distributions have similar shapes (Figs. 3 and 5) and are closely correlated in their central tendencies and spreads across all six lithologies (Fig. 4), consistent with our hypothesis that fracture spacing distributions can be used to predict initial particle size distributions in sediment.

Our talus size measurements do not, in contrast, support the null hypothesis that initial particle shape is invariant for blocks produced by mechanical weathering. At the 11 sites where we measured the a-, b-, and c-axes diameters, we quantified shape by calculating b:a and c:a ratios, which can be plotted together on a ternary diagram (Fig. 6) that displays rods, slabs, and equisided blocks at the vertices (Sneed and Folk, 1958; Graham and Midgley, 2000). At many of the sites, individual particles span nearly the full range of shapes represented in the diagram. Within each rock type there is little site-to-site variability in mean particle shape, suggesting that we can group sites together by rock type. When we do, we find statistically significant differences in mean particle shape among some rock types, despite substantial overlap in the distributions of individual shapes among the lithologies (Fig. 6). For example, talus produced from the metasediment have b:a and c:a ratios of 0.57 and 0.26. respectively, on average, and is therefore more elongate on average than talus produced from the granodiorite, which have corresponding ratios of 0.64 and 0.35 (Fig. 6a). In addition, andesite particles are more slab-like than basalt on average, with a lower mean c:a ratio of 0.26, compared to 0.36 (Fig. 6b). In both of these comparisons, many of the talus deposits have similar elevation and therefore similar climatic conditions (Table 1), suggesting that the differences in shape among the rock types are due to intrinsic differences in bedrock rather than differences in weathering conditions. Our results show that different rock types have different initial b:a and c:a ratios and thus conflict with the theoretical expectation that b:a and c:a ratios should have universal values of 0.67 and 0.45, respectively, for particles produced by mechanical weathering (Domokos et al., 2015). This suggests that lithology-specific values for initial shape may be needed when using shape to infer distance traveled from sediment sources (Miller et al., 2014; Szabo et al., 2015; Novak-Szabo et al., 2018), particularly for lithologies that have foliation and other anisotropic properties.

## 4 Discussion

The close correlations between talus size and fracture spacing distributions at our sites (Figs. 3–5) suggest that particles are detached by mechanisms that exploit most if not all of fractures exposed on the cliff faces and do not undergo much size reduction due to physical or chemical weathering in talus deposits. This finding, while limited to our sites, is robust across a wide range of lithologies and weathering conditions, suggesting that it spans a range of processes that could lead to particle detachment and subsequent weathering in talus deposits.

### 4.1 Predicting initial sizes from fracture spacing

Of the three particle dimensions measured here, the distribution of b-axis diameters most closely matches the fracture spacing distributions (Figs. 3–5). This suggests that fracture spacing measurements can be used to predict the initial size distribution of intermediate particle diameters. This is useful because the b-axis diameter is the most characteristic linear measure of particle volume and is therefore commonly used to represent particle mass in sediment transport theory and applications (Bagnold, 1966). Nevertheless, across our sites the b-axis is systematically $\sim\frac{1}{2}$ $\Phi$ interval larger on average than the median fracture spacing (Fig 4b), raising the question of whether this reflects a bias in our methods or incomplete exploitation of fractures during detachment from the cliff faces.

There are three potential sources of bias in our methods. First, the systematically coarser b-axis diameters may be partly driven by vertical sorting that causes fine particles to be underrepresented in point counts conducted on talus slope surfaces. However, we find no evidence of this in our measurements of sediment extracted from openings between surface particles: at each site where we made these measurements, the size distributions of the bulk samples overlap with the fine tail of the talus distribution enough that they could be combined using established techniques (Bunte and Abt, 2001) into a single continuous distribution. In this case, fine particles are sufficiently numerous on the talus surface that the statistics of the measured surface size distribution would not be affected by loss of a fraction of the fine-tail particles to the interstitial pores. This suggests that analysis of the talus at the surface provides unbiased estimates of the size distributions of material shed from cliff faces at our sites.

Another possible explanation for the systematic offset in b-axis size may stem from limitations inherent in one-dimensional measures of size to characterize three-dimensional objects. Fracture spacing is an indirect measure of latent block volume, and axis diameter is an indirect measure for talus volume. One alternate approach, used by Messenzehl et al. (2018), is to estimate talus particle volumes from linear measurements of talus axes and characterize fracture density in joints per cubic meter using scanline measurements of fracture spacing. However, there is no significant difference in the offset when we apply this approach to our data. Thus, we conclude that using linear rather than volumetric metrics is an unlikely source of the difference between median b-axis and fracture-spacing data.

A third possible methodological explanation stems from the random orientation of the scan lines relative to the joint sets
exposed on the rock face. If a scan line traces diagonal transects across a set of prismatic rectangular blocks, where the latent
a- and b- axes are exposed and the c-axes extend into the rock mass, then the measured fracture spacings would be
systematically larger than the typical b-axis, by a factor that depends on the angle between the scan line and the joint set making
up the b-axis fractures. However, if the a-axes extend into the rock mass, then the scan line measurements would instead
underestimate the b-axis spacings. In the most general case, blocks are formed by three intersecting joint sets with non-
perpendicular orientations, and thus are not rectangular prisms. In this case, the inter-fracture distance measured along any
given scan line crossing a block could possibly range from near zero (near the tip of an acute-angled point) to greater than the
a-axis length (for example if spanning the longest possible linear distance across a rectangular block face). An essential
assumption in using the scan-line technique is that this variability can be overcome by a large sample size, resulting in an
accurate if imprecise estimate of the central tendency and spread in the underlying population of fracture spacings (Priest and
Hudson, 1976; Roy et al., 2014). The close correspondence being particle sizes and spacings in all aspects of the distributions
except axis offset suggests that our sample sizes are large enough to overcome any scanline biases.
The fourth explanation is not related to measurement technique, but to the mechanics of block detachment. At many sites we
observe talus blocks that contain fractures that were not exploited by the detachment process (Fig. 2d, 2e). This could at least
partly explain the offset between measured median b-axis diameters and median fracture spacing, with detachment favoring
longer, more persistent fractures exposed on the cliff face. Incomplete exploitation of fractures could occur where longer, more
persistent fractures extend more rapidly than shorter fractures, intersecting to detach relatively large blocks.  The effect appears
to be most prominent at MT39, where even the c axes are coarser than the fracture spacing overall (Fig. 3b). We cannot rule
out the possibility that some of the unexploited fractures were created or extended by stresses during or after detachment from
the cliff and deposition on the talus slope. In any case, these unused fractures can be exploited during later size reduction by
physical and chemical weathering. Based on our observations, we conclude that incomplete exploitation of fractures provides
the best explanation for why b-axis diameters are systematically greater than fracture spacings across our sites.
The finding that the b-axes are systematically greater than the fracture spacings contrasts with previous measurements from
other mountain landscapes in California, where sediment sizes also correlate strongly with—but are systematically finer than—
fracture spacings in the source bedrock (Neely and DiBiase, 2020; Sklar et al., 2020). In the Inyo Creek catchment, for example,
the fine mode of the bimodal sediment size distribution on hillslopes is controlled by weathering processes that vary
independently of fracture spacings (Sklar et al., 2020). The coarse mode is also roughly 40% smaller than the fracture spacings,
potentially reflecting limited resolution of the photo-based fracture spacing measurements. This bias toward larger fracture
spacings may also help explain why particles are smaller on average than fracture spacings at the San Gabriel and San Jacinto
mountains sites, where photos were used to quantify fracture spacings (Neely and Dibiase, 2020). However, variations in

particle size across these sites can also be explained by differences in weathering that are driven by variations in the fractional coverage of regolith. In these steep mountain landscapes, the rough surfaces of bedrock hillslopes provide locations favorable to transient storage of coarse particles produced on adjacent slopes. During storage, particles are subject to physical weathering processes, such as frost cracking and thermal stresses, and chemical weathering processes aided by the presence of water and vegetation. In contrast, the relatively smooth and nearly vertical cliff faces at our study sites lack locations favorable to transient particle storage, and the adjacent talus surfaces are well-drained and minimally vegetated. Hence, we conclude that the offset toward finer sizes evident at the Inyo Creek, San Gabriel, and San Jacinto sites is due in part to substantial weathering not experienced on the bare cliff faces and adjacent talus slopes at our sites.

**4.2 Latent size versus weathering**

Our analysis suggests that there are offsets but no systematic site-to-site deviations in slope from the 1:1 trend between median b-axis diameters and median fracture spacings, despite the large differences in climate and thus weathering environment (Table 1). Moreover, there is no significant trend in residuals relative to the 1:1 trend with either mean annual temperature or average annual precipitation. This implies that the latent size distribution (embedded within the fractures exposed on the cliff faces) dominates over in-situ weathering as the main control on the particle sizes in talus cones across our sites.

The dominance of latent size over weathering is also supported by previous analyses of correlations between fracture density and erosion rates at the Sierra Nevada sites, where talus deposit volumes have accumulated since deglaciation ~13,000 years ago (Gillespie and Zehfuss, 2004) were used to quantify cliff retreat rates (Moore et al., 2009). Higher fracture density (and thus lower fracture spacing) corresponds to faster cliff retreat rates (Table 2), because denser fractures contribute to lower rock mass strength, which makes bedrock cliffs more susceptible to erosion (Howard and Selby, 2009; Moore et al., 2009; Glade et al., 2017; Eppes et al., 2018; Neely et al., 2019). Thus, at these sites, where weathering is minimal and cliffs are still responding to deglaciation, fracture spacing controls both initial size and the production rate of sediment through its effects on rock mass strength. In soil mantled landscapes, in contrast, where hillslope erosion rates are set by stream incision rates, theory and observations suggest that faster erosion should generally lead to larger initial particle sizes due to shorter residence times ($T_R$) during which rock might weather as it is exhumed through the fractured bedrock and saprolite portions of the critical zone (e.g. Attal et al., 2015; Riebe et al., 2017; Sklar et al., 2017; Callahan et al., 2019). For the purposes of understanding controls on initial particle size, we define residence time as

$$T_R = H_W / E \tag{1}$$

where $H_W$ represents the thickness of the subsurface zone where bedrock might weather during exhumation prior to initial particle detachment, and $E$ represents the mean rate of exhumation, equal to surface erosion rate at steady state. At our Sierra Nevada cliff-talus sites, and in general at other sites where $T_R \sim 0$, latent size should commonly dominate over weathering in setting initial particle size distributions.

### 4.3 Sediment production timescales

Our analysis of cliff retreat rates and fracture spacings from the Sierra Nevada points to another potentially insightful timescale: $T_P$, the time required to liberate a latent particle having the characteristic, median size, calculated as

$$T_P = F_{50}/E \tag{2}$$

where $F_{50}$ is the median fracture spacing, a proxy for the characteristic latent particle size, and $E$ is the erosion rate of the surface where particles are initially detached. At our Sierra Nevada sites, $E$ is equal to the rate of cliff retreat. Application of Equation 2 to data from our sites indicates that $T_P$ is as short as 88 years for detachment of one layer of 60-mm diameter latent particles from the cliff at EP-26, the most rapidly eroding cliff face. At the most slowly eroding cliff face, CB-1, Equation 2 suggests that it took 16,500 years to detach one layer of 330-mm diameter particles, indicating that the entire post-glacial accumulation time and more was needed to detach a volume equivalent to a single layer of latent particles with the characteristic median size (Table 2). The calculated $T_P$ at the remaining talus-cliff pairs in the Sierra Nevada sites is less than 13,000 years, consistent with the assumption in the cliff retreat rate calculations of Moore et al. (2009) that all of the sediment now contained within the talus piles was produced after the glaciers retreated.

The relative importance of latent size and weathering can be evaluated by quantifying the ratio of the residence time ($T_R$) to the particle production time scale ($T_P$),

$$T_R/T_P = \frac{H_W}{E} / \frac{F_{50}}{E} = H_W/F_{50} \tag{3}$$

Erosion rate cancels out of Equation 3 because both $T_R$ and $T_P$ are inversely proportional to the rate at which rock is converted to mobile particles. At our sites, which represent an extreme end member with no saprolite and minimally weathered bedrock (i.e., $H_W \sim 0$), $T_R/T_P << 1$, and the latent size distribution dominates over weathering in setting initial particle size. Although uncommon, this extreme may occur in diverse settings, including arid landscapes, glacial valleys, steep rapidly eroding mountains, and over the scale of individual hillslopes in soil mantled landscapes where bedrock exposures (and thus areas with $T_R \sim 0$) are patchy. In contrast, in soil mantled landscapes, where the thickness of saprolite and weathered bedrock is both large and spatially extensive, weathering should dominate over latent size distributions. This can produce the other extreme, $T_R/T_P >> 1$, particularly where erosion is also slow and fractures are closely spaced.

In between the two extremes, we envision a spectrum in the relative importance of weathering and latent size as a function of $T_R/T_P$. This spectrum is illustrated conceptually in Figure 7 for three idealized hillslopes with the same slope and erosion rate. Increasing saprolite and weathered bedrock thickness and decreasing fracture spacing should lead to higher $T_R/T_P$ ratios (from left to right in Fig. 7), which in turn would correspond to finer initial sediment size distributions produced at the top of fractured bedrock or saprolite (cf. Fig. 7a, b, and c). Fig. 7a depicts a case at the transition from bare bedrock (e.g., the cliff faces studied here), to slopes with patchy soil cover, such as those observed at the other California sites where fracture spacing and sediment size have been quantified (Neely and DiBiase, 2020; Sklar et al., 2020). Such sites should have enhanced potential for weathering relative to our sites. This might help explain why the median b-axis diameters plot higher than median fracture

spacings at our sites and vice versa at the other sites in California. In landscapes that are completely covered with regolith and
weathered rock (Fig. 7b–c), the signal of the latent size distribution (and also of initial shape; Fig. 6) may fade before liberation
of sediment into the soil. Thus, initial size should be dominated by weathering as residence times increase and thereby increase
the exposure of rock to chemical and physical weathering during exhumation through the critical zone (Fig. 7c).

**4.4 Implications for future work**
Our results from talus-cliff pairs show that, where bedrock is exposed, the latent sediment size observed in fracture spacing
distributions can be used to predict initial sediment size. Latent size should also influence initial sediment size in soil-mantled
landscapes, in combination with the influences of in-situ physical and chemical weathering, except in the extreme condition
of long and/or intense exposure to weathering. This suggests that more widespread fracture spacing measurements would
contribute improved understanding of controls on the evolution of sediment size in landscapes. However, these measurements
are difficult to obtain. In soil-mantled landscapes, direct measurements of fractures in bedrock can often be made where
relatively unweathered bedrock is exposed in roadcuts or in outcrops, but these measurements may be biased if outcrops are
not representative of nearby soil-mantled rock. Fracture spacing has also frequently been quantified from boreholes and cores,
but have only rarely been used in critical zone studies (e.g., Holbrook et al., 2019) because they have mostly been obtained for
different purposes (e.g., oil and gas exploration). Geophysical techniques can also be used to characterize subsurface fracture
density (e.g., St Clair et al., 2015), and when calibrated by direct observations from cores and outcrops (Flinchum et al., 2018;
Callahan et al., 2020), may provide estimates of absolute fracture spacing (Parsekian et al., 2015). Fracture density can also be
estimated from rock mechanical models (Shen et al., 2019) and based on analysis of topographic and regional stresses (Slim
et al., 2015; Moon et al., 2020). These techniques will be central to quantifying the interplay of fracturing and weathering in
determining initial sediment size using our time-scale-based conceptual framework.

Our finding of a strong association between rock type and fracture spacing, and thus initial sediment size, also suggests a
potentially fruitful avenue for future work. Our study was not designed to systematically test for the influence of rock type on
fracture spacing, but the topic has spawned a large literature that might be mined for general relationships relevant for
predicting hillslope sediment size (Narr and Suppe, 1991; Lore et al., 2001; Manda et al., 2008). Overall, our finding of a
strong link between fracture spacing and initial sediment size for the end-member case of talus-cliff pairs strengthens the
foundation for future field and modeling studies of the influence of lithology, tectonics, and climate on landscape-scale
variations in hillslope sediment size.
**5 Conclusions**
The detachment of rock fragments from fractured bedrock on hillslopes creates sediment with initial size distributions that set
the upper limits on particle size for all subsequent stages in the evolution of sediment as it is exposed to chemical and physical

weathering during transport from source to sink. The initial size distribution should depend on three main factors: the size distribution of latent sediment (i.e., blocks defined by throughgoing fractures); weathering that occurs in-situ, in fractured bedrock before the sediment is detached; and weathering and during the detachment process (e.g., disintegration along crystal grain boundaries). However, the initial size distribution is difficult to measure, because the interface across which sediment is produced is often shielded from view by overlying soil. Talus deposits that have accumulated beneath cliff faces offer opportunities to test the hypothesis that, when in-situ weathering is minimal, the initial size distribution should strongly reflect the latent size distribution defined by fractures on the cliff faces.

Here, we presented measurements of fracture spacing and particle size distributions from talus-cliff pairs spanning a wide range of climates and lithologies in California. Median fracture spacing varies by a factor of 40, median particle size varies by a factor of 60, and both of these variables correlate strongly with lithology. In addition, fracture spacing and talus size distributions are closely correlated with each other in central tendency, spread, and distribution shape, with b-axis diameters showing the closest correspondence with the fracture spacing at most sites. This suggests that weathering has not modified latent sediment, either before or during detachment from the cliff face. In addition, talus has not undergone much weathering after deposition and is slightly coarser than the latent sizes implied by the fractures, because the talus contains unexploited fractures inherited from the cliff face. Where detachment processes leave many of these unexploited fractures, initial sediment size and fracture spacing may not correlate as well as they do across our sites, even where in-situ weathering is minimal (e.g., Messenzehl et al., 2018). The positive offset in b-axis diameter relative to fracture spacing at all but 2 of our talus-cliff pairs differs from previous work elsewhere in California, where b-axis diameters are systematically finer than bedrock fracture spacings (Neely and DiBiase, 2020; Sklar et al., 2020), likely due to post-detachment weathering and possible biases in photo-based fracture spacing measurements. Together, these observations support a new conceptual framework illustrating the relative importance of latent size distributions and weathering on the initial sediment size distribution in mountain landscapes. In this framework, hillslopes occupy a spectrum defined by the ratio of two characteristic timescales: the residence time in saprolite and weathered bedrock, and the time required to detach the characteristic particle size. Where weathering residence times are negligible, as at our 15 talus-cliff pairs, the latent size distribution can be used to predict the initial size distribution. At the other end of the spectrum, where weathering residence times are long, the latent size distribution will provide limited predictive information about initial sediment size distributions.

**Author Contributions**

JV, LS, and JM designed the field investigation and JV and LS carried it out. JV, LS, and CR analyzed the data. LS and CR prepared the manuscript with contributions from all co-authors.

**Competing interests**
The authors declare that they have no conflict of interest
**Data availability**
All data are published and freely available at the URL contained in the following data citation:
SKLAR, LEONARD, 2020, "Verdian_Esurf_2020_Data", https://doi.org/10.5683/SP2/NYGA6T, Scholars Portal Dataverse,
V1, UNF:6:m1jd5WeIpA0s4SIe+NMavg== [fileUNF]
**Acknowledgments**
The authors thank Nolen Brown, Navek Ceja, Van Jackson-Weaver, Bradley Penner, Paul Vawter, and Dan Rosenburg for
assistance in the field; Russell Callahan for assistance with graphic design in Figure 1; John Caskey and Zan Stine for helpful
discussions. Comments by two anonymous reviewers and Associate Editor Simon Mudd helped improve the manuscript.
Funded by the National Science Foundation grants EAR 1324830 to Sklar and EAR 2012357 to Riebe. Additional support for
Verdian and Sklar provided by the Doris and David Dawdy Fund for Hydrological Science at San Francisco State University.

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

**Table 1. Study sites in Sierra Nevada and San Francisco Bay Area**

| Location | Pair | Lithology | Latitude (°N) | Longitude (°W) | Elevation (m) | MAT[1] (°C) | MAP[2] (mm) |
|---|---|---|---|---|---|---|---|
| Conness Basin | CB-1 | Granodiorite | 37.9763 | 119.3082 | 3262 | 1.3 | 1190 |
| Conness Basin | CB-2 | Granodiorite | 37.9750 | 119.3033 | 3293 | 1.1 | 1226 |
| Conness Basin | CB-3 | Granodiorite | 37.9797 | 119.3006 | 3171 | 1.5 | 1179 |
| Conness Basin | CB-5 | Metasediment | 37.9928 | 119.2870 | 3140 | 1.3 | 1152 |
| Ebbetts Pass | EP-24 | Andesite | 38.5655 | 119.8084 | 2530 | 4.8 | 1343 |
| Ebbetts Pass | EP-25 | Andesite | 38.5665 | 119.8114 | 2549 | 4.8 | 1343 |
| Ebbetts Pass | EP-26 | Basalt | 38.5473 | 119.8136 | 2732 | 3.9 | 1455 |
| Ebbetts Pass | EP-28 | Basalt | 38.5483 | 119.8144 | 2744 | 3.9 | 1455 |
| Mount Tallac | MT-38 | Granodiorite | 38.9430 | 120.1235 | 2134 | 6.5 | 1481 |
| Mount Tallac | MT-39 | Granodiorite | 38.9420 | 120.1247 | 2195 | 6.5 | 1481 |
| Grizzly Peak | GP-1 | Basalt | 37.8903 | 122.2346 | 393 | 13.8 | 727 |
| Twin Peaks | TP-1 | Pillow basalt | 37.7504 | 122.4483 | 260 | 13.6 | 705 |
| Twin Peaks | TP-2 | Pillow basalt | 37.7502 | 122.4476 | 252 | 13.6 | 705 |
| Twin Peaks | TP-3 | Chert | 37.7533 | 122.4480 | 280 | 13.6 | 705 |
| Twin Peaks | TP-4 | Chert | 37.7533 | 122.4480 | 280 | 13.6 | 705 |

[1]Mean annual temperature (Prism Climate Group, 2019)

[2]Mean annual precipitation (Prism Climate Group, 2019)

**Table 2. Results**

| Pair | Fracture spacing[1] (mm) | a-axis diameter[1] (mm) | b-axis diameter[1] (mm) | c-axis diameter[1] (mm) | Fracture geometric stdev | b-axis geometric stdev | Erosion rate (mm/yr) | $T_p$ (yrs) | Layers removed[3] |
|---|---|---|---|---|---|---|---|---|---|
| CB-1 | 330 | 375 | 250 | 130 | 0.346 | 0.357 | 0.02 | 125,000 | 0.8 |
| CB-2 | 200 | 500 | 270 | 205 | 0.458 | 0.612 | 0.09 | 3,000 | 5.9 |
| CB-3 | 280 | 720 | 420 | 190 | 0.551 | 0.477 | 0.05 | 8,400 | 2.3 |
| CB-5 | 120 | 335 | 220 | 95 | 0.309 | 0.306 | 0.25 | 880 | 27 |
| EP-24 | 80 | 225 | 150 | 55 | 0.283 | 0.255 | 0.31 | 484 | 50 |
| EP-25 | 155 | 320 | 200 | 70 | 0.401 | 0.294 | 0.12 | 1,667 | 10 |
| EP-26 | 60 | 95 | 55 | 25 | 0.285 | 0.201 | 0.68 | 81 | 150 |
| EP-28 | 70 | 160 | 100 | 65 | 0.358 | 0.338 | 0.26 | 385 | 48 |
| MT-38 | 390 | 1010 | 575 | 310 | 0.387 | 0.333 | 0.09 | 6,389 | 3.0 |
| MT-39 | 200 | 850 | 570 | 280 | 0.411 | 0.352 | 0.14 | 4,071 | 9.1 |
| GP-1 | 77 | 130 | 82 | 48 | 0.286 | 0.203 | - | - | - |
| TP-1 | 10 | - | 10 | - | 0.251 | 0.328 | - | - | - |
| TP-2 | 10 | - | 14 | - | 0.314 | 0.215 | - | - | - |
| TP-3 | 24 | - | 27 | - | 0.214 | 0.14 | - | - | - |
| TP-4 | 19 | - | 26 | - | 0.257 | 0.184 | - | - | - |

[1]Fracture spacings and particle diameters are reported as medians of distributions measured in field

[2]Cliff retreat rates were measured by Moore et al. (2009).

[3]Layers removed is the number of layers of thickness equal to the median fracture spacing that have been removed since the glacier retreated and is calculated as 13,000/$T_P$, where $T_P$ is calculated according to equation 1.

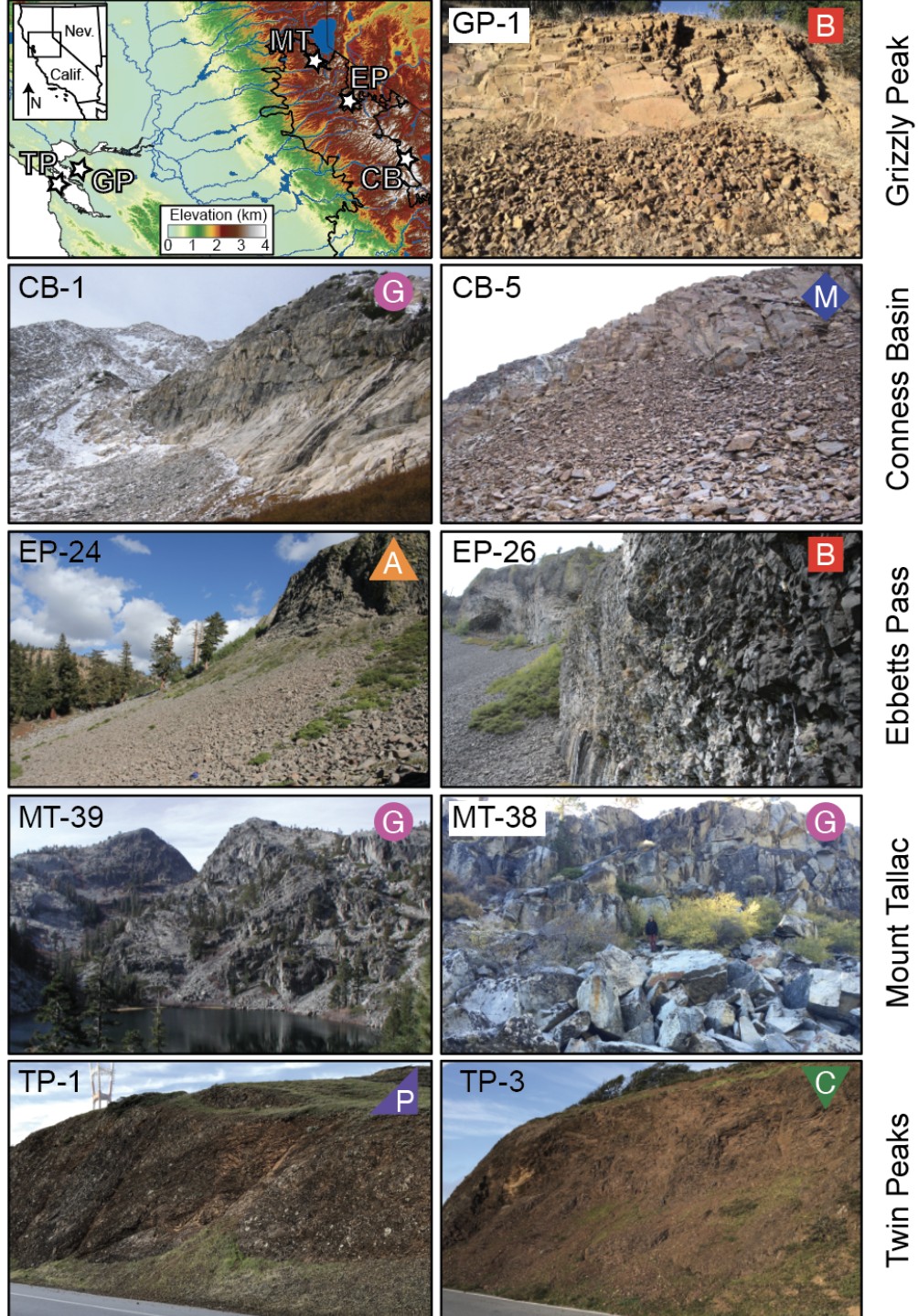

**Figure 1.** Study site map (upper left) and representative talus-cliff pairs from each site, with label designating lithology (red B squares = basalt; pink G circles = granodiorite; purple M diamond = metasediment; orange A triangles = andesite; purple P triangles = pillow basalt; green C triangles = chert). Scale varies between images. See Table 1 and text for site descriptions.

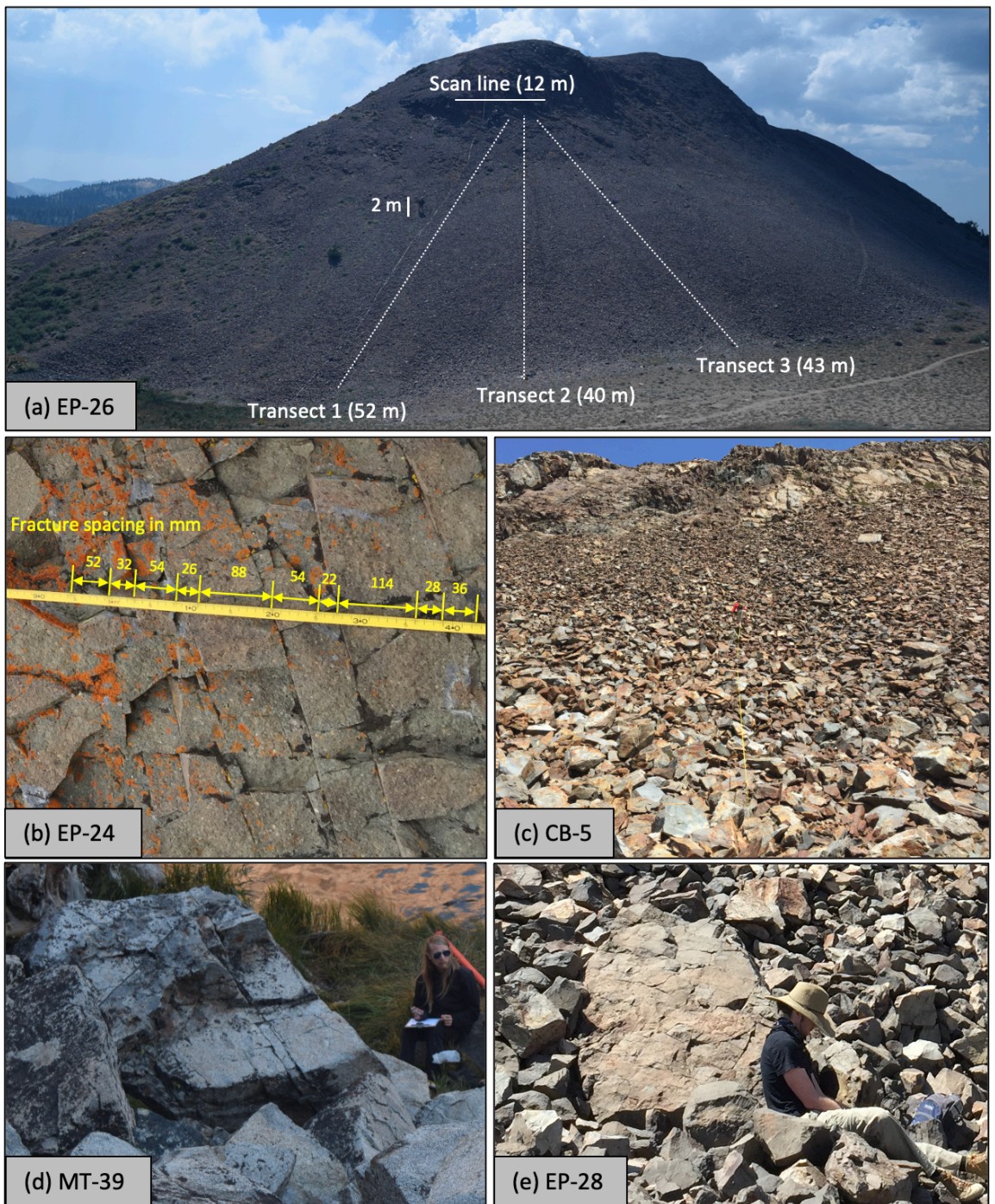

**Figure 2.** Field survey methods, showing: (a) an example of transect line and scan line layout at EP-26 site in the Sierra Nevada; (b) detail of scan line at EP24, illustrating measurement of distances between fractures along scan line; (c) establishment of transect 2 at CB-5; and talus boulders containing unexploited fractures at MT-39 (d) and EP-28 (e). 2 m scale in (a) highlights a person near transect 3.

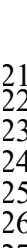

**Figure 3.** Particle size and fracture spacing distributions for each study site, sorted by median b-axis diameter in descending order from left to right and top to bottom. Colored lines show fracture spacing distributions, solid black lines show b-axis diameter distributions, dashed lines show a- and c-axis diameter distributions. (a-axis and c-axis diameters plot to the right of left the b-axis diameters respectively). Color codes and labels for lithology are as in Fig. 1; see Table 1 for site abbreviations. At the chert and pillow basalt sites, only b-axis diameters were measured (see text).

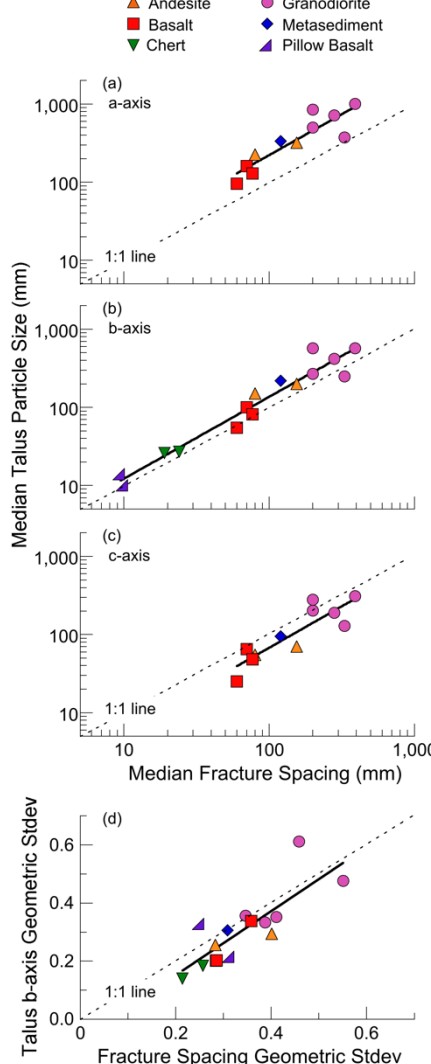

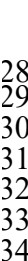

**Figure 4.** Central tendency and spread of particle size and fracture spacing distributions for median a-axis diameter (a),
median b-axis diameter (b), median c-axis diameter (c), and the geometric standard deviation of the b-axis diameters (d). In
each case, across the wide range in particle sizes and fracture spacings represented by the different lithologies sampled here,
there are strong correlations between the particle size and the fracture spacing distributions that are statistically
indistinguishable from a 1:1 relationship (a–d). The correspondence is closest for the b-axis diameters (b), though they are
systematically 42% larger on average than the fracture spacings.

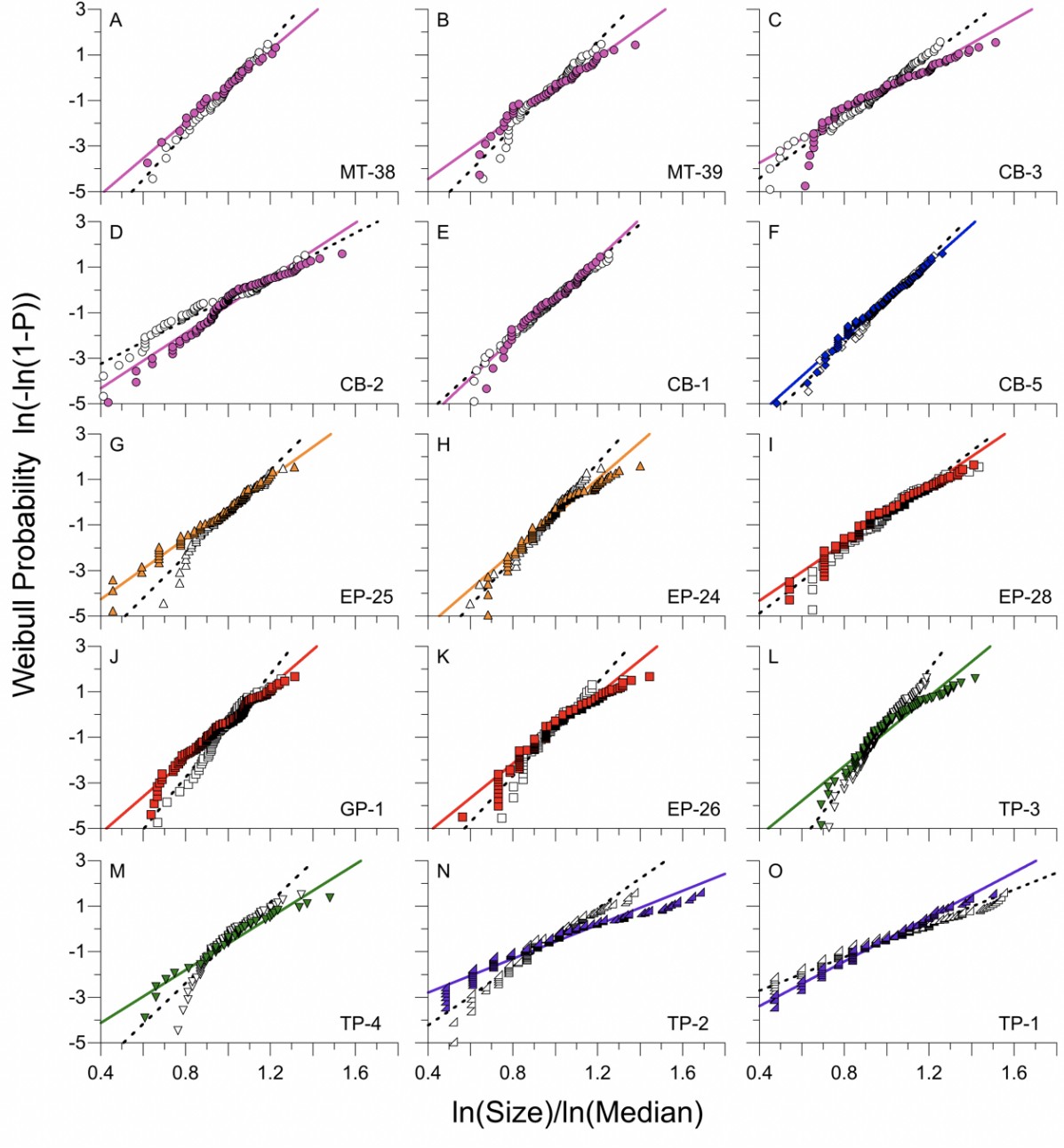

**Figure 5.** Comparison of distribution shape for the b-axis diameter (open symbols) and fracture spacing (colored symbols) distributions in Weibull probability space (see text). Points fall along a straight line in these plots when the sample is drawn from a population having a Weibull distribution. Best-fit linear regressions for b-axis diameters and fracture spacings are shown as dashed and colored lines, respectively. Colors correspond to lithologies following conventions in Fig. 1. For most sites, most points plot along a straight line, implying that their population distributions are consistent with a Wiebull distribution. In addition, the data commonly overlap, consistent with a close match between the shape of the particle size and fracture spacing distributions at many of the sites. Examples and exceptions are highlighted in the text.

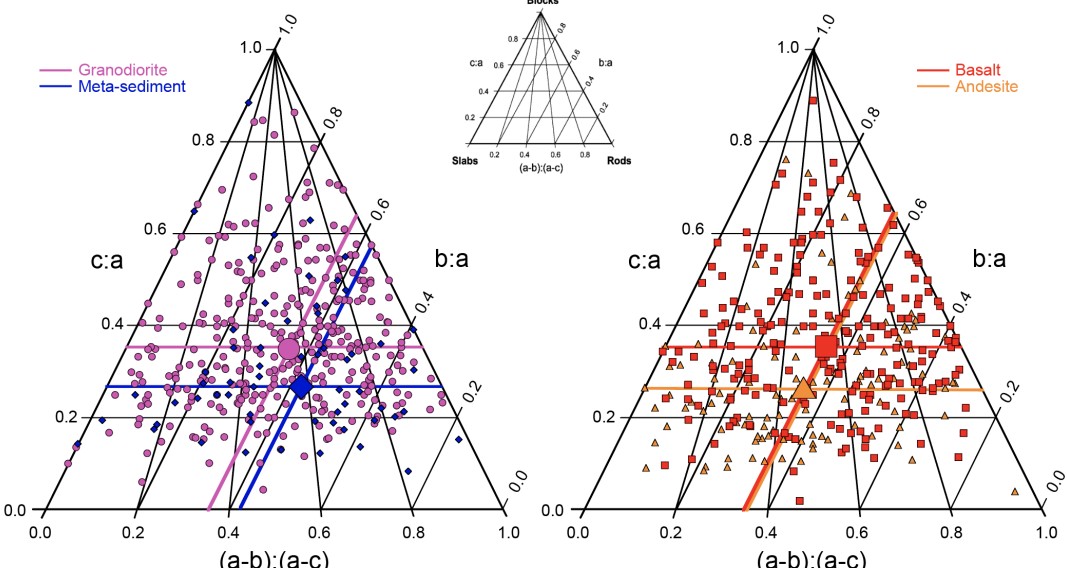

**Figure 6.** Initial particle shape at sites where all three particle diameters were measured, as revealed in ternary diagrams with blocks, rods, and slabs at vertices (inset). Although data within each site and within each lithology are widely scattered in shape, the central tendencies for samples grouped by lithology yield several statistically significant differences. For example, granodiorite has a higher b:a and c:a ratio than the metasediment (left), indicating that metasediment is more rod-like on average. Symbols and colors represent lithology following conventions of Fig. 1.

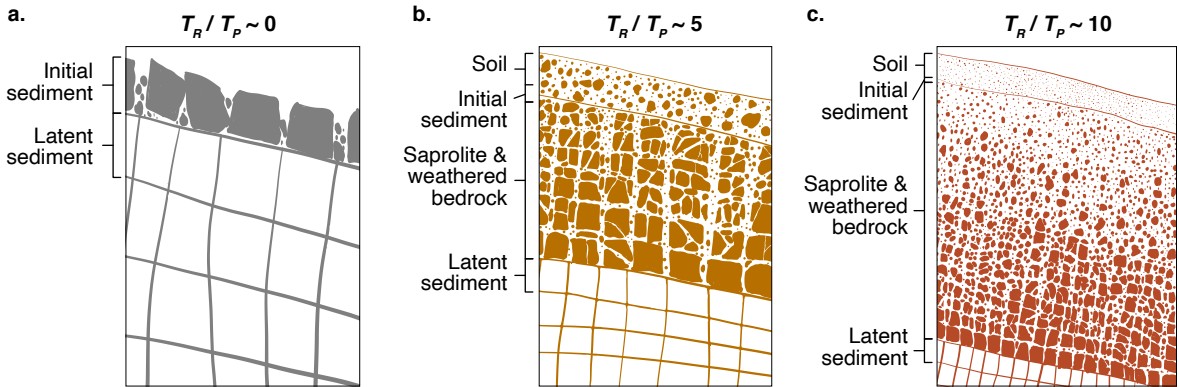

**Figure 7.** Conceptual framework illustrating how initial sediment size is influenced by latent sediment size and weathering. Panels depict idealized vertical profiles of subsurface weathering where initial sediment is produced by detachment at the top of intact bedrock (a) or saprolite (b, c). Erosion rate is the same in each panel. Where fracture spacing is wide and where the weathering zone is thin, latent size should dominate over weathering (a), and vice versa where fractures are closely spaced and the weathering zone is comparatively thick (c). These examples lie on a spectrum of outcomes that correlate with the ratio of two characteristic timescales: the timescale of weathering ($T_R$) and the time required to detach a layer of characteristic (median) particles at the base of mobile regolith ($T_P$). Higher ratios correspond to a greater influence of weathering and a lesser influence of latent size on initial sediment size distributions.