# Peer review of "Sediment size on talus slopes correlates with fracture spacing on 1"

_Earth Surface Dynamics, 2020_

## Referee Comment (RC1) · Anonymous Referee #1 · 4 Aug 2020

The problem of predicting sediment size distributions on hillslopes has seen recent interest in the geomorphology community. Sediment size, especially when larger than typical "soil" grain sizes, influences dynamics both on hillslopes and adjoining river channels that must carry hillslope-derived sediment. However, data regarding hillslope sediment size and dynamics is scarce. Important parameters such as initial sediment size distributions, as well as their subsequent evolution through transport and weathering through the hillslope system, are not well known. A common assumption is that fracture spacings can be used to estimate initial size distributions in hillslope sediment

derived from bedrock outcrops. However, weathering of blocks in situ or immediately after detachment may alter the initial size distribution.

In order to test this, Verdian et. al present field measurements of fracture spacing in exposed rock walls and sediment size in immediately adjoining talus slopes at 5 different sites in California. They find that size distributions are not smaller than their respective rock wall fracture spacings (in fact, they are coarser) and conclude that weathering does not substantially alter the initial size distribution. However, they find that initial sizes and to some extent particle shape do depend on lithology. Finally, they propose a simple ratio of timescales that determines the importance of weathering in setting size distributions before detachment from parent material (bedrock or saprolite).

Given the scarcity of hillslope sediment size data, the data presented in this paper are useful for their own sake. The basic confirmation that fracture spacing sets initial size distributions is also useful for models of hillslope sediment transport. However, I find the scope of the paper and presentation and interpretation of the data to be misleading and confusing. Below I outline the main points that need to be addressed, along with suggestions for improvement.

Main suggestions

The primary claim of the paper is that pre-existing fractures in exposed bedrock cliffs set the initial size distributions of hillslope sediment. This is in contrast to the idea that weathering- either in situ or soon after sediment detachment- substantially alters sediment sizes. It is currently unclear in the paper 1) why we care about latent fracture spacing 2) how "initial" sediment size is defined 3) why initial sediment size in important.

On one hand, I appreciate that the authors have taken the time to try to test this hypothesis with field data. The sediment size and fracture spacing data are vitally important for our understanding of hillslopes. On the other hand, it seems fairly obvious that boulders in a talus pile immediately next to a rock wall would have sediment sizes that correspond to the fracture spacing in that rock wall. Fracture spacings are measured at

**ESurfD**
the free boundary of the wall, where weathering processes can occur. Isn't it possible that the measured fracture spacings reflect a combination of latent and weathering-derived fractures? Either way, what is the functional importance of grain size set by latent fractures vs. a combination of latent and weathering? It is perhaps more interesting to find that talus material has not weathered since release from the wall. However, because there is no constraint on age of the talus pile, it is difficult to draw conclusions about the relevant weathering timescale. Further, the data presented in this paper do not have any relevance for subsequent weathering of sediment as it moves through the hillslope system and ultimately into channels. Other studies have found sediment size fining indicating a combination of selective transport and weathering. The authors contrast their findings with Neely and Dibiase 2020 and Sklar et al 2020, but these studies measured sediment sizes far away from their exposed bedrock sources. The authors should clarify that their findings have no bearing on sediment size evolution and transport long after detachment.

Looking at Figures 3 and 4, I am not convinced that sediment sizes in the talus pile are indistinguishable from fracture spacing in the rock wall. In fact, they seem to be substantially larger at many of the sites. The authors explain that this may because block detachments occur along wider fracture spacings, and subsequent blocks thus contain some of the smaller fractures within them. This is interesting and a fine interpretation, but conflicts with the conclusion that fracture spacing can be used to predict initial sediment size on hillslopes. The authors point out that p values for their data are large enough to be "insignificant." However, it is unclear how p values are calculated and whether they are meaningful for the data presented: "In each case, the increase in median particle diameter with increasing fracture spacing follows a trend with a slope that is statistically indistinguishable (p>0.45) from a 1:1 relationship in log-log space." From figure 4 it looks like perhaps the slopes are equivalent, but particle sizes are substantially shifted from fracture spacing. The smallest offset between particle size and fracture spacing is 42%. Let me be clear: this is not a flaw in the data, but in the interpretation and presentation.

Finally, it would be very helpful for the authors to more clearly outline the importance of this data. Even if fracture spacings perfectly matched sediment size in the talus piles, how is this helpful for future studies? Fracture spacing is very difficult to measure accurately, even in landscapes where clear, exposed bedrock exists. It is even more difficult in soil-mantled landscapes. The framework proposed in the paper is useful, if only because it points toward the difficulty and necessity for us to better understand rock and saprolite weathering and its role in producing initial sediment sizes. Perhaps one of the most interesting findings in this paper is the difference in grain size between lithologies. I think the paper would be much more interesting and useful if the authors amplified these findings. Overall, I think the findings in this paper are useful for advancing our fundamental understanding of hillslopes: but this does not come across clearly in the paper. I hope the authors can clarify the importance of their work.

Minor points

Sediment shape: Figure 6 shows differences in sediment shape between lithologies. While this is useful information, the discussion around shape could be toned down. The authors contrast their findings with Domokos et al., 2015, stating that "there is no evidence in our data that initial particle shape varies with size, contrary to the predictions from previous work that smaller particles should be more block-like on average." However, the particle shape-size trends in Domokos et al., 2015 saturate for grains around 50mm in length. Most of the data in this paper are at or above this range, so I wouldn't expect them to see the shape-size trends. Perhaps a more interesting comparison is to look at differences between the average shape data presented here and the saturation values in Domokos ($\sim$0.425 for c:a and 0.675 for b:a). Further, I'm not sure that finding difference in mean shape values between lithologies can be directly compared with the probability distributions of shape parameters in Domokos et al.

Ratio of timescales: The authors propose a framework in which the ratio between regolith residence time and particle detachment time determine initial sediment sizes. I think this framework is fine; however, the way it is presented assumes that regolith is

necessary for weathering. I am still unconvinced that fracture spacing does not include effects from mechanical weathering (frost cracking, tree roots, thermal fluctuations, etc.). Perhaps the authors just need to clarify when the particle size clock starts (see next suggestion below). If it starts after particles are released from bedrock, then their framework makes sense. However if it starts when pure bedrock first begins to weather/crack, it may not be appropriate.

Definition of "latent" and "initial": It's currently a bit difficult to understand what the authors mean by "latent" and "initial." A clear definition in the introduction would help a lot.

Talus sampling: In line 139 the authors explain that spatially uniform sediment sampling along the talus slope should yield an accurate grain size distribution even with size-selective entrainment. However, this assumes that size distributions change linearly downslope. The authors might want to simply point out this assumption.

Figure 3: The authors refer to parts of the figure by letter, but they're not included in the figure.

Figure 4: A legend that identifies data points is needed. It is unnecessarily difficult to refer to figure 1 and remember the colors and shapes to understand figure 4.

Missing references: The authors should also cite some missing recent relevant studies: Shobe et al., 2016, who explore hillslope sediment size controls on river incision; Glade et al., 2017, who show data for boulder size distributions in an exposed bedrock system; and Glade and Anderson, 2018, who discuss the implications of weathering vs. erosion rate timescales on hillslopes; Ward 2019, who discusses ratios between incision rate and cliff retreat timescales; Duszynski et al., 2019 who review scarp retreat mechanisms and the role of weathering.

Suggested references:

Duszyński, Filip, Piotr Migoń, and Mateusz C. Strzelecki. "Escarpment retreat in sedimentary tablelands and cuesta landscapes–Landforms, mechanisms and patterns." Earth-Science Reviews 196 (2019): 102890.

Glade, R. C., and R. S. Anderson. "Quasi-steady evolution of hillslopes in layered landscapes: An analytic approach." Journal of Geophysical Research: Earth Surface 123.1 (2018): 26-45.

Glade, Rachel C., Robert S. Anderson, and Gregory E. Tucker. "Block-controlled hillslope form and persistence of topography in rocky landscapes." Geology 45.4 (2017): 311-314.

Shobe, Charles M., Gregory E. Tucker, and Robert S. Anderson. "Hillslope-derived blocks retard river incision." Geophysical Research Letters 43.10 (2016): 5070-5078.

Ward, Dylan J. "Dip, layer spacing, and incision rate controls on the formation of strike valleys, cuestas, and cliffbands in heterogeneous stratigraphy." Lithosphere 11.5 (2019): 697-707.

---

## Referee Comment (RC2) · Anonymous Referee #2 · 4 Aug 2020

Verdian et al., present a timely new dataset that measures bedrock fracture spacing on cliffs and compares the spacing of fractures to the grain size of sediment contributed to downstream hillslopes. The topic is very pertinent. Growing research shows that sediment grain size can impact relationships between topography and sediment transport at a range of spatial scales, and the need is growing for empirical datasets that constrain the initial grain size of sediment produced from fresh rock. This study primarily tackles the 2nd point above by collecting field data from a number of cliff locations previously studied by Moore et al., 2009. I found the connection between bedrock fracture

spacing and talus-sediment grain size convincing and promising, but have suggestions to hopefully strengthen the manuscript's clarity, impact, and context.

After some moderate revisions that address these comments, I would be happy to work with these authors towards publishing this manuscript in esurf.

Review aspects:

Does the paper address relevant scientific questions within the scope of ESurf? Yes. Does the paper present novel concepts, ideas, tools, or data? Yes. New data. Are substantial conclusions reached? Yes. Are the scientific methods and assumptions valid and clearly outlined? Could use clarification Are the results sufficient to support the interpretations and conclusions? Yes Is the description of experiments and calculations sufficiently complete and precise to allow their reproduction by fellow scientists (traceability of results)? Generally, yes. Do the authors give proper credit to related work and clearly indicate their own new/original contribution? Yes. Does the title clearly reflect the contents of the paper? Yes. Does the abstract provide a concise and complete summary? Generally, yes. Is the overall presentation well-structured and clear? Yes with suggestions Is the language fluent and precise? Yes but could improve Are mathematical formulae, symbols, abbreviations, and units correctly defined and used? Yes Should any parts of the paper (text, formulae, figures, tables) be clarified, reduced, combined, or eliminated? No Are the number and quality of references appropriate? Yes with some additions Is the amount and quality of supplementary material appropriate? Yes

Principle criteria

Scientific significance: good – ideas and concepts have existed and have been explored in other publications, but this study contributes useful new field datasets – particularly fracture spacing measurements paired with talus grain shape and size measurements. New conceptual model introduced in discussion section requires some clarification, connection to hillslope/catchment-scale variability in weathering zone thickness,

and specific factors that can be measured in the field.

Scientific quality: good – individual data collection methods are valid, but comparison between datasets and connection to pre-existing work should be clarified. Clarify quantitative methods used to distinguish results and make comparison between datasets. Datasets are very useful, but additional context is likely required to communicate results and interpretation to readers. I would like to see some more detail in the methods section.

Presentation quality: good/fair – Clear statement of testable hypothesis. I recommend adding to the background/study site section to frame tectonic, climatic, and lithologic context. Some organizational suggestions to distinguish results from interpretations (results section vs discussion section). Encouraged to add subheadings to discussion section and results section to improve organization of results and interpretations. Some additional context is needed to communicate the importance of the problem and the utility of the findings.

Line by line edits:

Abstract

Line 19: Before introducing results (median fracture spacing and particle size...), add a sentence describing methods used to measure fracture spacing (scan lines) and sediment grain size (a-b-c axis measurements). Potentially this could be incorporated into the previous sentence.

Line 19: Here and throughout the manuscript, consider switching "particle" with grain or clast? I would interpret particle to include wood or other debris that does not originate on cliffs. Just my preference though.

Line 20: Is there a metric you can provide to quantitatively describe the correspondence between fracture spacing and b-axis diameter?

Line 22: "weathering has not modified latent sediment either before" seems confusing

to me. You are measuring bedrock fracture spacing on the cliff faces (which are affected by near-surface weathering to some degree), so you haven't quantified how the latent sediment size changes before reaching the cliff face. It is important to define (possibly in the abstract but definitely in the introduction) at what point in rock exhumation to the surface you're measuring latent sediment size (see later comment)

Line 22: I would remove point about modification of sediment before detachment and focus on point about modification of sediment during detachment: i.e. grain size does not fine significantly during rockfall or clast spallation from the cliff face

Line 24: replace "it" with clasts? - "clasts contain some fractures inherited from bedrock" – is there any field evidence of this and is it possible to present this evidence (even if photographs of deposits)

Line 27-30: Possibly you can shorten these sentences and allow space to expand on your methods earlier in the abstract. Also, it's important to acknowledge landscapes between the two end-members mentioned, as many steep catchments fall into this domain space (some cliffs, but still some soil-mantled hillslopes).

Introduction

Line 33: I might reword "the life of sediment" and remove this opening phrase: (The size distribution of sediment influences chemical, physical, and . . . ect)

Line 36: Replace "The first stage begins" with (Initial sediment grain size is set when clasts are detached from. . . ect..)

Line 38: rephrase?: "The resulting initial size distribution is the starting point for the evolution of the size distribution of sediment on hillslopes, and therefore sets the upper limits on sediment size distribution as sediment is routed through the catchment network" ? – slightly more specific than the life-cycle of sediment

Line 41: good statement of overall problem that this paper is addressing

Line 46: Can you cite this point? Or add a qualifier to the beginning of this sentence: "We hypothesize that. . ." or "The "latent" size distribution of clasts is thought to be set by . . . (add a citation to the Sklar et al., 2017 paper maybe if no others come to mind?)

Line 48: I feel like this paragraph should likely be split into at least 2 if not 3 or 4 paragraphs. Here might be a place to do it.

Some place in this paragraph you need to specify where in the rock exhumation timeline you are considering fractured rock to reflect a latent clast size.

It probably makes sense to define the time to measure "latent" sediment size at the point when "clasts" are at the cliff face (this is where you are measuring fracture spacing), but it's important to note that near-surface processes that fracture bedrock can modify fracture spacing before rock reaches the cliff face surface where your fracture spacing measurements are occurring. The true "latent" clast size reflecting the first interaction between the topographic surface and fractures likely occurs at some depth beneath the cliff face, where stress induced by non-flat topography becomes non-negligible (i.e. Miller and Dunne, 1996 – https://doi.org/10.1029/96JB02531) or temperature fluctuations can fracture rock due to frost cracking (Hales and Roering, 2007 – https://doi.org/10.1029/2006JF000616 ) or diurnal heating and cooling (harder to get specific depths below surface where this process is important?. . . Collins and Stock, 2016 – rockfall triggered by thermal cycling, or work by Epps: Epps and Keanini, 2017?).

I think it is good to pick cliff surfaces as the surface where "latent" clast sizes are quantified (because this is most possible to measure), but it's important to clarify that modification of clast sizes related to near-surface processes may have potentially started at deeper depths, and the amount of subsurface latent clast size modification before reaching a cliff face might depend on climate, hillslope relief, or tectonic stress orientations.

Line 48: If you describe the point above at the end of the 3rd sentence in the current
paragraph, I think it makes sense to split the next sentence into a new paragraph (describing processes that fracture rock unrelated to the near surface – tectonic fracturing, unloading, and cooling)

Line 54: I would split the sentence starting with "As weathering commences" into a new paragraph that talks about near-surface processes that fracture bedrock. (climate, topographic stress perturbations, deep weathering)

Line 60: New paragraph starting with "Although these hypotheses.." End this new paragraph re-iterating that you are defining the "latent" sediment size as the spacing of fractures on exposed cliff faces?

Line 61-62: I believe this point has been explored by Messenzehl et al., 2018 and Neely and DiBiase, 2020 (this is in preprint.. so I'm not sure if you can cite this contribution yet?, but I see this is cited elsewhere due to relevance). I would reword to say: "the relative importance of latent sizes and weathering in initial size distributions has been explored systematically across a limited suite of climatic and lithologic settings" ?

Line 66-67: maybe re-iterate at the end of this sentence that prior studies focused on a limited range of bedrock lithology and climate variables?

Line 70: It might be important to note the resolution of the fracture spacing surveys in this study (could only resolve fractures with apertures > xxx cm). Your fracture spacing measurements occur at much higher resolution.

Line 73: Same comment as above (could only resolve fractures with apertures > 1-3 cm).

Line 73: change DiBiasi to DiBiase

Line 74: Results from Neely and DiBiase show that latent clast sizes dominate on steep slopes where bedrock cliffs are exposed, but not on gentle soil-mantled slopes.

Line 74: May add reference and background to Attal et al., 2015

(https://doi.org/10.5194/esurf-3-201-2015 ) and Roda-Boluda et al., 2018 (https://doi.org/10.1002/esp.4281 ), which look at erosion rate and hillslope morphology controls on sediment grain size. These studies do not quantify bedrock fracture spacing, but their results imply a link between weathering on soil mantled slopes and sediment grain size.

Line 76: I like contrasting the Messenzehl findings with the prior findings from California landscapes. Maybe move the following sentence (lines 78-79) up to the end of this paragraph to state how your study fits in with these prior investigations?

Line 81-82: change "initial sizes" and "latent sizes" to "talus clast sizes" and "latent clast sizes" ?

Line 85: Also, would mention that clast detachment could occur along pre-existing fractures that are below your detection limit. The small-aperture fractures may also be pre-existing.

Line 87: Another reason talus grain size may not match fracture spacing is if talus sediment is sorted after detachment. This is mentioned later in the manuscript, but should be mentioned here as well.

Line 87: new paragraph at "Neither of these alternatives"?

Methods

Line 98: Before the methods section, I think readers need a "study site" section that describes the various tectonic, climatic, and lithologic settings of each site. (Analyzing cliff/talus systems across these different variables is a main strength of the paper!). This section should at a minimum:

- Detail the location and tectonic setting/history of the outcrops (how does this connect to the inherited bedrock fracture network? (maybe add a map figure to show this too).. are some outcrops closer to active or inactive (– dead) faults?

[Figure]

- Details the climate and climate history of outcrops – relation to frost-cracking? diurnal heating and cooling? Some of this information is in table 1, but the climate history may also be important.

- Introduces the prior work used from Moore et al., 2009 (how did Moore et al., 2009 estimate cliff retreat rates used in later parts of the manuscript?)

Line 103: Include a source after the statement that says three sites where differences in lithology correspond to differences in average fracture spacing?

Lines 108 – 120: Scan lines have been used to measure bedrock fracture spacing in a number of applications, and it would probably be good to cite some of the studies that developed/used these methodologies.

Lines 108-120: One of my main critiques of this study is that the comparison is not straightforward between sediment grain size along an A-B-C axis and fracture spacing on a scanline. This needs to be stated clearly, because this impacts how to interpret results presented from the study.

Scanlines the way they are described (to my knowledge in this manuscript) do not usually run through the longest or shortest axis of a fracture-bound block on a cliff. A horizontal scanline will likely be skew across the fracture-bound block unless all fractures are perfectly vertical. The spacing between the fractures on the horizontal scanline is usually not the A-B or C axis of the fracture-bound block. See Figure 2 in the manuscript, lower right.

While the scanline fracture spacing is still a useful measure of fracture spacing, I'm not surprised that the B-axis of the clasts is larger than the scanline-fracture spacing in the results, because the scanline-fracture spacing likely does not sample the B-axis of the clast (scanline crosses corners of clasts and does not go along the widest or shortest axis).

For example, the Neely and DiBiase study in preprint measured fracture spacing as

the short-axis of fracture-bound blocks on orthophotos of cliff faces. This assumes that the short-axis of fracture-bound blocks on the cliffs represents the B-axis. Though this isn't true for "latent" particles where the visible short-axis of the fracture-bound block is actually the C-axis, measurement of the short axis of the fracture-bound block might be more straightforward to map to the B-axis of a detached clast (see figure attached).

The difference between these methods needs to be clarified for the reader here, and this should be mentioned when interpreting the results (unless I'm missing something about how fracture spacing was measured)?

(See attached figure)

If scaled photographs exist for part or all of the scanlines (i.e. figure 2 lower left), another option would be to measure the short and long axis of clasts along the scanline on the photographs (with a photo resolution limit of 2mm) (i.e. attached figure right panel). This could be compared to the scanline fracture spacing and the A-B-C axis of the clasts in the talus piles. This could be a useful comparison for future studies connecting fracture spacing to geomorphic processes or sediment grain size.

Line 124: Slope-parallel transects result in 300 clasts measured per talus pile? Correct? Maybe state this if true?

Line 124: again I'm preferential to saying clasts instead of particles (because particles may include wood or really anything that doesn't necessarily come from the cliffs)

Line 128: "everything else" – I'd replace this with "larger clasts"

Line 129: to be clear, "particles" are "talus cone clasts"? Also I'm confused, if the fracture spacing resolution limit was 2 mm, how do you compare talus clasts <2 mm to the "latent" clast size, which cuts off at a detection limit of 2 mm (presumably) ?

Lines 134-139: I agree with this section, but how do you know that the talus cone captures all of the sediment grain size distribution spalled from the cliff? ... the largest clasts roll the furthest, do some of these traverse the whole talus pile? If the talus pile

is mined by an active stream at its base, the largest clasts may be somewhere else downstream and the talus pile undersamples the coarsest clasts.

It's been shown in a couple studies/settings that the coarsest sediment grain size distribution is typically found at the base of colluvial/headwater channels in steep landscapes: Hack 1965 https://pubs.usgs.gov/pp/0484/report.pdf), Brummer and Montgomery, 2003, https://doi.org/10.1029/2003WR001981 , and Neely and DiBiase preprint figure 9 https://www.essoar.org/pdfjs/10.1002/essoar.10502617.2

It might be important to note this limitation when comparing bedrock fracture spacing to the grain size of sediment on talus cones. But maybe these talus piles act as better 'clast traps' than angle of repose talus cones...

Also, it looks like the bottom two sites in figure 1 have roads at the bottom of them? I'm not sure if large clasts can travel far enough from the cliffs to reach these roads, but these clasts would likely be cleared by road crews?

Line 140-145: These methods are mentioned, but I don't see the results presented anywhere. Or the methods detailing how these results are integrated into the full distributions are missing? Does this particle size distribution replace the fine tail of the talus grain size distribution that is below the resolution limit of the fractures (2 mm?). Do you quantify fracture spacings finer than 2 mm? even though this is the detection limit for the fractures?

Lines 147 – 151: This section seems out of place. Maybe move this to be near the other paragraphs that describe how you measure sediment grain size (1st paragraph in this section?)

Methods section suggestions overall:

- Need to clarify difference between scanline fracture spacing and a-b-c axis measurement of clast/fracture-bound block

- A number of statistical techniques are later mentioned in the results section. It would

help me to introduce these techniques in the methods section and explain how these techniques are used to quantitatively distinguish between the different distributions and test the hypothesis of the paper?

- Detail how the "fines" are accounted for. Where are the bulk sample sediment results presented? How are these included into the distribution? Is the same thing performed with the fracture spacings?

- Note that talus cones may not capture the largest grains if the base is actively mined by a steep and competent stream (or a road maybe?.. not sure)

Results

Line 155: "c-axis diameters as small as 2 mm" – what about the bulk sediment samples (fines?)

Line 161: cumulative empirical distribution function is not a jargon-y term? May be helpful to define in the methods section?

Lines 159-168: A lot of this comparison is qualitative. I see more quantitative comparison in the next paragraph, but I feel unprepared to understand this comparison because the methods section did not introduce how the quantitative comparison would occur.

Line 167: I would like to see the discussion section return to the anomalous result from MT-39 – do the clasts here have many fractures still retained ? – if the fracture spacing is smaller than the c-axis even?

Line 173: I'm not quite sure how these "p" values were calculated or what they mean. I see some comparison to a 1:1 correlation and I think I know how you could do this, but in the methods section, could this be clarified and related to how you are testing your hypothesis?

Line 178 – 181: this is an interpretation and could be moved into a discussion section?

– I am a fan of splitting the results and discussion sections.

Line 185-186: Would be helpful to clarify how p-values are calculated in methods section

Lines 193-194: Would be helpful to clarify in the methods section how the Weibull distribution is populated/used to compare with your data? I find this hard to interpret without some background on the methods.

Line 194-195: Change "the degree to which the data follow a Weibull distribution at each site is illustrated in Figure 5" to a figure call to figure 5 at the end of the previous sentence "(Fig. 5)"

Lines 190-210: Be clear to specify that "A,B,C-axis" distributions come from talus sediment and not from fracture-bound blocks on cliffs. It's not clear to me how the Weibull distributions are quantitatively linked to the results presented? Most of the comparison seems qualitative at this point? Is there a way to quantify this comparison like with the p-values? If so this should be reported and introduced in the methods section.

Lines 200 and 202: "In some cases" – is there anything specific about these cases? Should these be discussed more in the discussion section?

Lines 217: reword? "Within each rock type, there is little site-to-site variability in mean particle shape. When sites are grouped by rock type, we find . . .."

Line 220: how much lower are the mean b:a, c:a ratios?

Line 221-223: this is an interpretation and could be moved to a discussion section?

Lines 229 – 234: this point also reads like a discussion point (contextualizing with prior work) – reorganize?

Discussion:

Overall – I had a hard time following parts of the discussion. Potentially, points should
be organized into subheadings?

- Significance/reliability of comparison between scan line fracture spacing and sediment grain size in talus piles?

- Comparison with observations from other landscapes where bedrock fracture spacing was quantified?

- Reasons for differences between fracture spacing and sediment grain size? (sorting, kinetic sieving, incomplete breaking along fracture planes, measurement resolution?)

- Climate/weathering controls on initial sediment grain size?

- Conceptual model (with some revisions/clarification?)

Line 237: "nearly the full network of fractures" .. Rephrase: "from the network of fractures with apertures >2 mm"?

Lines 237-238: "This finding" and "it" are hard to unpack in this sentence. I'm having a hard time interpreting this sentence.

Lines 241 – 247: This might be a good place to return to the discussion about the difference between scan-line fracture spacing and the long and short axis of a fracture-bound block. Note that Neely and DiBiase (check spelling) uses the short axis of fracture-bound blocks, which may account for the difference between these study results.

Line 249: See above comment, also Neely and DiBiase –preprint - and Sklar et al., 2020 also have coarser fracture detection limits? (1-3cm in Neely and DiBiase (typically 2 cm) and not sure about Sklar et al., 2020 because a variety of techniques are used – aerial imagery likely has a coarser detection limit though?). The fracture spacing will increase with coarser fracture detection limits – but maybe fracture scaling relationships could be used to compare between these studies – probably a non-trivial endeavor (see Hooker et al., 2014 and similar work that quantifies the distribution of fracture

apertures in rock masses https://doi.org/10.1130/B30945.1 ) Either way, I would make sure to acknowledge the difference in technique used to measure fracture spacing and the difference in minimum resolvable fracture aperture when comparing these results to results from these other two studies.

Lines 251-252: Not sure where these results are presented??

Line 257: are there field photographs of this? These would be useful to show if even qualitative?

Lines 261-265: this reads more like a results section. I think it would be useful to incorporate the climate data comparison with fracture spacing/ talus grain size in some sort of figure? Plot mean annual temp & precip. Vs median fracture spacing/talus B-axis ? Same thing with lithology vs median fracture spacing/talus B-axis? This would show any correlation or lack of correlation between these variables and fining the sediment after it spalls off of the cliffs? Again, I think the study sites ranging across various climates and lithologies is a strength of this paper and should be used/analyzed! – though some more analysis might need to be done to correctly synthesize these additional parameters/forcings.

Lines 269-272: Background on the analysis by Moore et al., 2009 should be introduced in the beginning of the manuscript. Also, I had a hard time following the organization of this paragraph. Maybe split paragraph into discussion of cliff-dominated systems Tr ~0, and soil-mantled systems Tr » Tp?

Lines 275-276: Add reference to Attal et al., 2015 and Roda-Boluda et al., 2018

Line 276: a bit unclear what is meant by "sites", yes if the site is only a talus and cliff system, the weathering zone is very thin or only along fractures and Tr = 0, but there is a problem if you scale this up to a full catchment (which is relevant for landscape evolution), full catchments typically include a mix of bare-bedrock and soil-mantled hillslopes (see DiBiase et al., 2012 https://doi.org/10.1002/esp.3205 ; Milodowski et al.,

2015 www.earth-surf-dynam-discuss.net/3/371/2015/doi:10.5194/esurfd-3-371-2015 ;
Neely et al., 2019 https://doi.org/10.1016/j.epsl.2019.06.011 )

Line 276: I'm struggling to understand the spatial scale over which Tr ∼0. At the catchment scale, catchments are rarely 100% bare bedrock (there's still some soil). At what point does the catchment system switch from Tr = 0 to Tr » Tp?

Lines 279 – 311: I think it would be useful to define the spatial scale over which these timescales (Tr and Tp) are being assessed.

Consider breaking this up into paragraphs that address this issue at the cliff/hillslope scale, and then the catchment scale (which mixes sediment from cliffs and soil-mantled hillslopes)?

At the scale of a single cliff or hillslope, I think this ratio/analysis may be helpful, but at the scale of a steep catchment with cliffs, usually hillslopes are a mix of soil-mantled slopes, talus slopes, and bare-bedrock cliffs. It becomes challenging to define 'H' in these settings or ensure that 'E' is the same for soil-mantled slopes and cliffs. (See Neely et al., 2019 https://doi.org/10.1016/j.epsl.2019.06.011 and Neely and DiBiase preprint https://doi.org/10.1002/essoar.10502617.2 and Sklar et al., 2020 https://doi.org/10.1002/esp.4849 )

I do think this is a useful discussion point to link the study to past work, but maybe this should be reframed?

- i.e. at the scale of exposed cliffs and downslope talus, Tr « Tp and sediment grain size reflects latent clast sizes. The emergence of cliffs in steep landscapes then implies a larger supply of sediment with grain sizes that mirror bedrock fracture spacing?

Also, it's not clear to me how long the sediment in the talus cones remains in the talus cones? Tr could be very long on the talus cones, but lack of biota, water retention, and soil could lower the efficacy of weathering on these cones. So the sediment is detached, sits in the weathering zone as mobile regolith, but there is little fining occur-

ring?

For example, post-glacial rock piles and boulder fields may not develop soil for tens to hundreds of thousands of years despite being in temperate climates on low slopes (see Denn et al., 2017 https://par.nsf.gov/servlets/purl/10098651)

In the boulder field setting, Tr » Tp, but the latent particle size still dominates? – maybe this is a special case, but relict boulder fields are common in post-glacial landscapes such as the ones maybe studied by Moore et al., 2009??

Line 305: transition to patchy soil cover in these landscapes may be better described by DiBiase et al., 2012 https://doi.org/10.1002/esp.3205 , Heimsath et al., 2012 http://www.nature.com/doifinder/10.1038/ngeo1380 , and Neely et al., 2019 https://doi.org/10.1016/j.epsl.2019.06.011

Line 305: not sure where these patchy-soil cover landscapes fit in the context of figure 7, because some of the hillslopes within these landscapes are Fig. 7a, and some are Fig. 7b. Also change "DiBiasi" to "DiBiase"

Line 306-307: I'm not sure about this point with the difference in fracture detection resolution and fracture spacing measurement technique (scanline vs short-axis between fracture bound block).

Lines 308-311: I think this point has been demonstrated in other analyses: Attal et al., 2015; Roda-Boluda et al., 2018; Neely and DiBiase preprint. The data from this study does not really show much soil weathering (unless some data is replot to show sediment fining in the talus piles due to climate variables??), so maybe this point should be rephrased, cited, or removed?

Line 483: why 13000? Is there are source/data suggesting this is when deglaciation occurred near these cliffs? Should this number be the same for each cliff?

Figure 1: I think it would be useful to expand the study map figure into its own figure (new figure 1) and show the geologic context (bedrock lithology and distribution of

active and inactive faults). Can you add a scale bar to each panel? I'm concerned about the road in the bottom two panels, and I'm not sure where in panel CB-1 and MT-39 the measurements occurred (highlight the transects a bit?)

Figure 2: add a scale bar to bottom right image? Could use this figure to illustrate how the scan-line fracture spacings were measured?

Figure 4: add p-values referenced in text that compare to the 1:1 line to panels?

Figure 5: mark resolution detection limits on x-axis of panels?

Figure 6: I find the C:A notation a bit confusing and prefer C/A, B/A, A-B/A-C notation (just my preference though). Nice figure!

Figure 7: I'm a bit confused about the scale/dimensions. Make sure to have (A) (B) and (C) markers on panels?

I'm not sure about the assumption that the erosion rate and slope are the same despite different fracture spacing or weathering zone thickness (in caption and in discussion section).

For example, soil transport rate for a given slope may change in response to different clast size (larger blocks require steeper slopes to move at same erosion rate – see Neely et al., 2019 https://doi.org/10.1016/j.epsl.2019.06.011 , DiBiase et al., 2018 https://doi.org/10.1016/j.epsl.2018.10.005 ) If climate changes drive difference in weathering zone thickness, that may also change the hillslope erosion rate – see Owen et al., 2011 https://doi.org/10.1002/esp.2083 )

I'm not sure the assumptions in this conceptual model are realistic because weathering zone thickness is often coupled to hillslope erosion rate, fracture spacing, and climatic/ecologic factors?

— These are a lot of edits! But I think most of these items can be addressed or clarified? The dataset is a valuable foundation, and I think with some more context and

specific language surrounding the methods, this will be a very useful study that can be pulled into a lot of future research! Hopefully this isn't discouraging or too intimidating.

**ESurfD**
[Figure]

[Figure]

**Fig. 1.**

---

## Author Comment (AC1) · 4 Jan 2021

Verdian et al. Anonymous Referee #1

Referee Comment (overview): The problem of predicting sediment size distributions on hillslopes has seen recent interest in the geomorphology community. Sediment

size, especially when larger than typical "soil" grain sizes, influences dynamics both on hillslopes and adjoining river channels that must carry hillslope-derived sediment. However, data regarding hillslope sediment size and dynamics is scarce. Important parameters such as initial sediment size distributions, as well as their subsequent evolution through transport and weathering through the hillslope system, are not well known. A common assumption is that fracture spacings can be used to estimate initial size distributions in hillslope sediment derived from bedrock outcrops. However, weathering of blocks in situ or immediately after detachment may alter the initial size distribution.

In order to test this, Verdian et. al present field measurements of fracture spacing in exposed rock walls and sediment size in immediately adjoining talus slopes at 5 different sites in California. They find that size distributions are not smaller than their respective rock wall fracture spacings (in fact, they are coarser) and conclude that weathering does not substantially alter the initial size distribution. However, they find that initial sizes and to some extent particle shape do depend on lithology. Finally, they propose a simple ratio of timescales that determines the importance of weathering in setting size distributions before detachment from parent material (bedrock or saprolite).

Given the scarcity of hillslope sediment size data, the data presented in this paper are useful for their own sake. The basic confirmation that fracture spacing sets initial size distributions is also useful for models of hillslope sediment transport. However, I find the scope of the paper and presentation and interpretation of the data to be misleading and confusing. Below I outline the main points that need to be addressed, along with suggestions for improvement.

Author Answer: We are grateful to Referee #1 for these thoughtful and constructive comments and suggestions. They are very useful for highlighting where we can improve our analyses, interpretations, and explanations. In our responses below, and associated changes to the manuscript, we have done our best to address each comment and concern.

Main suggestions

Referee Comment 1: The primary claim of the paper is that pre-existing fractures in exposed bedrock cliffs set the initial size distributions of hillslope sediment. This is in contrast to the idea that weathering- either in situ or soon after sediment detachment- substantially alters sediment sizes. It is currently unclear in the paper 1) why we care about latent fracture spacing 2) how "initial" sediment size is defined 3) why initial sediment size in important.

Author Answer: This is a helpful comment because it shows that the original manuscript did not successfully communicate several essential elements of the work.

1) We care about fracture spacing because intersecting fractures create a set of blocks that represent a "latent" or potential set of sediment particles (note: fracture spacing is not latent, the sediment particles are). The term latent is useful to distinguish between the blocks within the bedrock and the sediment particles they could become once they are fully detached from the bedrock. The latent particle size distribution, set by the fracture spacing distribution, sets the size distribution of sediment particles when they are first detached from bedrock and entrained in the geomorphic transport system, in situations where bedrock has not been subject to substantial weathering prior to detachment. In those situations, the initial sediment size distribution can then be estimated directly from knowledge of bedrock fracture density, or predicted indirectly from the lithologic, tectonic, and topographic factors that influence the extent of bedrock fracturing. Where weathering not insignificant, the initial sediment size distribution will reflect both sets of factors, those that determine latent size and those, such as climate, minerology, and erosion rate, that determine the extent of rock weathering prior to particle detachment. Predicting initial sediment size thus requires understanding the controls on both latent size and subsequent modification by weathering, and the relative importance of each in a given geomorphic setting.

2) "Initial" sediment size is defined as the size of particles when they are first (or initially) detached, or "released" from bedrock, and entrained in the geomorphic transport system. Sediment production is another term used for the concept that there is a threshold that separates intact rock from mobile sediment. When a sediment particle is first "produced", it has a size; that is the initial size.

3) The initial size distribution is important for at least two key reasons. First, the initial size sets the upper limit for sediment size in a catchment; in the absence of flocculation or cementation, sediment particles can only become smaller. Second, the initial size sets the scale for particle size reduction by comminution processes during transport. Particle size reduction in transport is commonly modeled as a function of three factors: the initial size, a transport distance (or time in transport), and a rate constant. For example, the Sternberg equation for particle size reduction in fluvial transport can be written as D(x) = Do*exp(a*x) where Do is the initial size, x is transport distance, and a is the rate constant. Thus, initial size influences particle size throughout a catchment. Environments that produce relatively larger initial size distributions (for example, due to less fractured rock), should have relatively larger sediment size distributions throughout the catchment, all else equal. To the extent that sediment size influences other geomorphic processes and landform attributes (e.g. river incision into bedrock and channel slope), initial size will be an important contributing factor.

Changes to the manuscript: We have substantially rewritten the introduction to more clearly articulate these three main points.

Referee Comment 2: On one hand, I appreciate that the authors have taken the time to try to test this hypothesis with field data. The sediment size and fracture spacing data are vitally important for our understanding of hillslopes. On the other hand, it seems fairly obvious that boulders in a talus pile immediately next to a rock wall would have sediment sizes that correspond to the fracture spacing in that rock wall.

Author Answer: As we stated in the original manuscript, the ("fairly obvious") expectation of a correspondence between fracture spacing in bedrock cliffs and talus size

on adjacent slopes is both intuitively appealing and mechanistically based, yet has not been systematically tested across a range of lithologies and weathering environments. Two recent studies in relatively uniform rock types (Neely and DiBiase 2020; Sklar et al., 2020) have shown a correspondence between fracture spacing and the coarse mode of hillslope sediment size distributions. However, another recent study by Messenzehl et al. (2018) showed that this correspondence does not always occur. In their case talus size is roughly uniform across a suite of locations where fracture spacing ranges over an order of magnitude; importantly, measured talus size is always either larger than or approximately equal to the mean fracture spacing. Their data and interpretation support the hypothesis that talus production by frost-cracking can preferentially exploit a subset of pre-existing fractures with a characteristic spacing, presumably leaving other fractures intact within the blocks detached from the cliff wall. These two conflicting sets of results provide additional motivation for our study, which extends the available data across a more diverse set of rock types and wider range of fracture spacings.

Changes to the manuscript: In the revised introduction we provide more detail on the results of Messenzehl et al. (2018), and explain how they provide an alternative hypothesis of control on talus size by a physical weathering process.

Referee Comment 3: Fracture spacings are measured at the free boundary of the wall, where weathering processes can occur. Isn't it possible that the measured fracture spacings reflect a combination of latent and weathering derived fractures?

Author Answer: (As noted above, it is not the fractures that we consider to be latent, it is the fracture-bound blocks that have the potential to become sediment particles when detached.) Yes, physical weathering processes could create new fractures within rock exposed at the cliff face, which would be included in our measurements of fracture spacing. By making these measurements at sites where rapidly retreating cliff faces expose relatively unweathered rock, we sought to minimize, rather than eliminate, the contributions of weathering to the fracture spacing distributions. However, in these settings, we can reasonably assume that weathering has at most a minor influence,

and that, if available, measurements of fracture spacing taken from drill cores at these sites would roughly reproduce the factor of 40 variation in median fracture spacing we observe at the rock face.

In our conceptual framework, we seek to distinguish the influence on the initial sediment size distribution of fracture-bound blocks, present in relatively unweathered rock at the base of the weathered zone, from the influence of weathering processes that can shift the initial sediment size distribution away from that of the fracture-block size distribution. Examples of such weathering processes include frost-cracking, as in the study by Messenzehl et al. (2018), and selective mineral dissolution that leads to disaggregation along crystal boundaries, which can produce a sand- or pea-gravel-sized initial distribution from crystalline rock that has a wide fracture spacing. Chemical weathering processes can also pre-condition rock by weakening it, such that it will produce initial sediment particles with sizes that reflect the scale of erosional detachment processes, such as the spacing between tree roots for sediment production by tree-throw or the size of animal limbs for sediment production by burrowing mammals. This study focuses on the end-member case where the influence of weathering is most likely to be limited to processes of detachment, rather than processes that significantly modify the rock at depth.

Changes to the manuscript: In the revised introduction we clearly define "latent size distribution" as the size distribution of fracture-bound blocks that would become sediment particles if particles are produced by detachment along those pre-existing fractures. We also clarify the distinction between weathering processes that contribute to the fracture spacing distribution at the surface, and weathering processes that have the effect of altering the initial sediment size distribution without affecting apparent fracture spacing.

Referee Comment 4: Either way, what is the functional importance of grain size set by latent fractures vs. a combination of latent and weathering?

Author Answer: Distinguishing the controls on initial sediment size is important for both understanding and modeling variations in sediment size across catchments. Whether initial size is controlled by fracture spacing acquired at depth, or by surficial weathering processes that tend to shift the initial size away from the fracture-determined latent size, has important implications for which landscape-scale boundary conditions are ultimately determining initial sediment size. Key controls on fracture spacing include rock strength and diagenetic origin, tectonic history, and topographic position. In contrast, weathering intensity and style will depend largely on climate, rock mineralogy, and erosion rate through its effect on residence time in the weathering zone. Our study is a small contribution to the larger problem of determining under what conditions fracture spacing is the dominant control.

Change to the manuscript: In the revised introduction we clarify this aspect of the study motivation.

Referee Comment 5: It is perhaps more interesting to find that talus material has not weathered since release from the wall. However, because there is no constraint on age of the talus pile, it is difficult to draw conclusions about the relevant weathering timescale. Further, the data presented in this paper do not have any relevance for subsequent weathering of sediment as it moves through the hillslope system and ultimately into channels. Other studies have found sediment size fining indicating a combination of selective transport and weathering. The authors contrast their findings with Neely and Dibiase 2020 and Sklar et al 2020, but these studies measured sediment sizes far away from their exposed bedrock sources. The authors should clarify that their findings have no bearing on sediment size evolution and transport long after detachment.

Author Answer: The Referee correctly notes that we did not measure the sizes of sediment other than on the talus slopes beneath the cliffs where sediment particles were produced, and thus do not present data that can be used to better understand the controls on rates of particle size reduction in transport long after detachment. However, we disagree with the statement that our measurements of initial sediment size have

"no bearing" on sediment size evolution. As noted above, initial sediment size sets the scale for subsequent size reduction. Furthermore, initial size affects subsequent transport dynamics, which in turn can affect rates of size reduction.

Changes to the manuscript: We have revised the paragraph in the discussion where we compare our results with those of Neely and DiBiase (2020) and Sklar et al. (2020), to explicitly consider the potential role of size reduction in transport in explaining the differences in the offset between sediment size and fracture spacing among the three studies.

Referee Comment 6: Looking at Figures 3 and 4, I am not convinced that sediment sizes in the talus pile are indistinguishable from fracture spacing in the rock wall. In fact, they seem to be substantially larger at many of the sites. The authors explain that this may because block detachments occur along wider fracture spacings, and subsequent blocks thus contain some of the smaller fractures within them. This is interesting and a fine interpretation, but conflicts with the conclusion that fracture spacing can be used to predict initial sediment size on hillslopes. The authors point out that p values for their data are large enough to be "insignificant." However, it is unclear how p values are calculated and whether they are meaningful for the data presented: "In each case, the increase in median particle diameter with increasing fracture spacing follows a trend with a slope that is statistically indistinguishable ($p>0.45$) from a 1:1 relationship in log-log space." From figure 4 it looks like perhaps the slopes are equivalent, but particle sizes are substantially shifted from fracture spacing. The smallest offset between particle size and fracture spacing is 42%. Let me be clear: this is not a flaw in the data, but in the interpretation and presentation.

Author Answer: This is a helpful comment because it shows where the original manuscript lacked clarity regarding the inferences that can be drawn from the data. We did not make a blanket claim that talus particle sizes are statistically indistinguishable from fracture spacing. Rather, we used the term "indistinguishable" for specific, narrowly-focused hypothesis tests using the data.

First, it may be helpful to remind the Referee and potential readers that data do not have p-values, hypothesis tests do. The passage quoted in this comment describes our test of the null hypothesis that the degree of correspondence between fracture spacing and talus size does not vary with the magnitude of fracture spacing. In plain language, this hypothesis holds that where fracture spacing is relatively wide, talus size should be relatively large, and where spacing is relatively close, talus should be commensurately smaller. Unlike the previous work of Neely and DiBiase (2020) and Sklar et al. (2020), we can use our data to test this hypothesis because we made measurements at sites that encompass a wide range of fracture spacings. This is the same null hypothesis that was implicitly rejected by Messenzehl et al. (2018), because talus size did not vary in parallel with fracture spacing at their sites. One way to test this hypothesis is to use linear regression. Because fracture spacing and talus size vary over more than an order of magnitude from site to site, we log-transformed the median values of both variables. We then fit linear trends (for each particle axis) to the log-transformed data using ordinary least squares regression. This analysis produces best-fit estimates of the trend-line slopes and their associated confidence intervals. We would reject the null hypothesis if the trends deviated significantly from a 1-to-1 relationship. This is a two-tailed, one-sample test because the slope of the 1:1 line is exactly 1.0, it has no uncertainty associated with it, and we make no assumptions about whether the talus trend might steeper or less steep than 1.0. In this case, the p-value quantifies the probability that our median talus-size measurements were sampled from a population with an underlying 1-to-1 correspondence with fracture spacing. We would reject the null hypothesis if that probability were sufficiently low. However, for the three regressions (one for each axis), the lowest p-value was 0.45 (the other two were higher), far greater than the conventional threshold of 0.05. From this result, we make the inference that the correspondence we observe, on average, between fracture spacing and talus size, does not vary meaningfully with fracture spacing; it is scale invariant and therefore may also occur in other settings with fracture spacings that differ from our study sites. We use a similar statistical analysis to test for differences

between talus size and fracture spacing in the spread of the distributions for each cliff-pair. We find that where the fracture spacing distribution has a narrow spread, so does the corresponding talus size distribution. More formally, we fail to reject (with p > 0.67) the null hypothesis that the spread in the talus sizes has a 1-to-1 scaling with the spread in fracture spacing across the range of fracture spacings measured. The results of these two hypothesis tests, taken together, provide strong support for the inference that the distributions of talus sizes at our sites are strongly influenced by the distributions of fracture spacings in the source rock. The corollary is that other factors, such as weathering processes that might impose a different distribution of initial particle sizes, are much less influential.

A second, and distinct issue, is the relationship between fracture spacing and the size of the three different talus particle axes measured. Even when fracture spacing and talus size are clearly correlated, the median fracture spacing may not match the median of any of the three axes, for a variety of possible reasons. In our data, the intermediate axis diameter comes closest, as we would expect, but is larger than the median fracture spacing, on average. Although this is clearly apparent in Figure 4, panel b, we report (in the Discussion) the results of a simple sign-test which rejects the null hypothesis that they are not different (with p = 0.006). Other statistical tests produce similar results: a paired-sample, 2-tailed t-test would reject the null hypothesis (with p = 0.003), as would the corresponding non-parametric Wilcoxon signed-rank test (p = 0.002). We certainly do not make the claim that the talus b-axis distributions are indistinguishable from the fracture spacing distributions.

One possible explanation for the systematic offset of b-axis size and fracture spacing may stem from the inherent limitations of using one-dimensional measures of size to characterize three-dimensional objects. Fracture spacing is an indirect measure of latent block volume, and axis diameter is an indirect measure for talus volume. Perhaps for this reason, Messenzehl et al. (2018) used their linear measurements of talus axes to estimate talus particle volumes; and used vertical and horizontal scanline measurements of fracture spacing to characterize fracture density in terms of joints per cubic meter of bedrock. However, when we convert our data in a similar manner, we obtain an almost identical result to the b-axis regression shown in Figure 4b. Thus, we conclude that the use of linear rather than volumetric metrics is unlikely to be the source of the difference between median b-axis and fracture-spacing data.

Another possible explanation stems from the random orientation of the scan lines with respect to the joint sets exposed on the rock face. If a scan line traces diagonal transects across a set of prismatic rectangular blocks, where the latent a- and b- axes are exposed and the c-axes extend into the rock mass, then the measured fracture spacings would be systematically larger than the typical b-axis, by a factor that depends on the angle between the scan line and the joint set making up the b-axis fractures. However, if instead it is the a-axis that extends into the rock mass, then the scan line measurements would underestimate the b-axis. In the most general case, blocks are formed by three intersecting joint sets with non-perpendicular orientations, and thus are not rectangular prisms. In this case, the inter-fracture distance measured along any given scan line crossing a block could possibly range from near zero (near the tip of an acute-angled point) to greater than the a-axis length (for example if spanning the longest possible linear distance across a rectangular block face). An essential assumption in using the scan-line technique is that this variability can be overcome by a large sample size, resulting in an accurate if imprecise estimate of the central tendency and spread in the underlying population of fracture spacings. However, biased estimates can result at any single site, for the reasons outlined above.

A third possible explanation is not related to the geometry of measurement technique, but to mechanics of block detachment. As the referee notes, we interpret the offset between measured median b-axis and median fracture spacing to be at least partly due to incomplete exploitation of the full set of fractures by the rock detachment processes. This would result in some talus particles, particularly the larger ones, retaining some of the more closely-spaced fractures measured in the bedrock cliff face. This interpretation is consistent with our observations in the field of fractured talus boulders, although some of those fractures could have been created or extended by stresses arising from the process of detachment from the cliff and deposition on the talus slope.

Finally, there is the question of how to predict initial sediment size. In environments where initial sediment size is dominantly controlled by the size of fracture-bound blocks at the surface of the source rock, relationships based on measurements or estimates of fracture spacing are likely to be a useful approach to predicting initial size. For example, if one assumes that our results can be generalized, one can write the following expression for the median b-axis (D50) as a function of the median fracture spacing measured by scan-line technique (F50): D50 = 1.42*F50, where the prefactor 1.42 accounts for the vertical offset of the b-axis regression line from the 1-to-1 line in Figure 4b. Whether this relationship accurately predicts talus particle sizes on other talus slopes beneath actively eroding bedrock cliffs could be explored with additional data. This simple expression serves to illustrate that fracture spacing can be used to predict talus particle size, in the case where there is a systematic offset to a scale-invariant correspondence between spacing and size.

Changes to the manuscript: In response to this comment, we have made a number of changes to the manuscript, including: - in the results section we expanded the explanation of the hypothesis tests applied to the data plotted in Figure 4; - in figure 2 we added a field photo showing talus boulders with fractures that can be interpreted as being inherited from the fractured bedrock of the cliff face above; - in the discussion section, we expanded the consideration of alternative explanations for the offset between the b-axis and fracture spacing; - in the discussion section, we added a paragraph, with additional relevant citations, discussing how measurements and model-based estimates of fracture spacing can be used to predict initial sediment size.

Referee Comment 7: Finally, it would be very helpful for the authors to more clearly outline the importance of this data. Even if fracture spacings perfectly matched sediment size in the talus piles, how is this helpful for future studies? Fracture spacing is

very difficult to measure accurately, even in landscapes where clear, exposed bedrock exists. It is even more difficult in soil-mantled landscapes. The framework proposed in the paper is useful, if only because it points toward the difficulty and necessity for us to better understand rock and saprolite weathering and its role in producing initial sediment sizes. Perhaps one of the most interesting findings in this paper is the difference in grain size between lithologies. I think the paper would be much more interesting and useful if the authors amplified these findings. Overall, I think the findings in this paper are useful for advancing our fundamental understanding of hillslopes: but this does not come across clearly in the paper. I hope the authors can clarify the importance of their work.

Author Answer: We agree that it is vital to clearly communicate the relevance of our results for future studies. Regarding the feasibility of obtaining fracture spacing measurements to predict initial sediment size, fracture spacing can be measured or estimated in a variety of ways. In soil-mantled landscapes, direct measurements of fracturing in bedrock can often be made where relatively unweathered bedrock is exposed in roadcuts or in outcrops such as along incising streams. Fracture spacing at depth can also be quantified using data obtained from drill cores, and a large literature exists reporting such measurements. Geophysical measurement techniques are also useful for characterizing sub-surface fracture density, and when calibrated by direct observations from cores and outcrops, may provide estimates of absolute fracture spacing. Fracture density can also be estimated from rock mechanical models, based on analysis of topographic and regional stresses. Application of these techniques will be important for exploring the utility of our time-scale-based conceptual framework for understanding the relative influences of fracturing and weathering in determining initial sediment size.

Our finding of a strong association between rock type and fracture spacing, and thus initial sediment size, also suggests a potentially fruitful avenue for future work. Our study was not designed to systematically test for the influence of rock type on fracture spacing, however a large literature exists on this broad topic, which might be mined for

general relationships relevant for predicting hillslope sediment size. Overall, our results confirming the expected correspondence of fracture spacing and initial sediment size, for the end-member case of bedrock cliffs producing talus-sized sediment, provides a stronger foundation for future field and modeling studies that seek to understand the influence of tectonics, lithology and climate in controlling landscape-scale variation in hillslope sediment size.

Changes to the manuscript: We have expanded the discussion (and added several relevant citations) to more directly address the implications of this work for future studies, focusing particularly on approaches for quantifying fracture spacing in soil mantled landscapes and the potential to use variation in rock type as a proxy for differences in fracture spacing and thus latent sediment size.

Minor points

Referee Comment 8: Sediment shape: Figure 6 shows differences in sediment shape between lithologies. While this is useful information, the discussion around shape could be toned down. The authors contrast their findings with Domokos et al., 2015, stating that "there is no evidence in our data that initial particle shape varies with size, contrary to the predictions from previous work that smaller particles should be more block-like on average." However, the particle shape-size trends in Domokos et al., 2015 saturate for grains around 50mm in length. Most of the data in this paper are at or above this range, so I wouldn't expect them to see the shape-size trends. Perhaps a more interesting comparison is to look at differences between the average shape data presented here and the saturation values in Domokos (_0.425 for c:a and 0.675 for b:a). Further, I'm not sure that finding difference in mean shape values between lithologies can be directly compared with the probability distributions of shape parameters in Domokos et al.

Author answer: We agree that it is difficult to compare our data with those reported by Domokos et al. (2015), in part because of incomplete reporting of the methods

they used to obtain their results. We accept the suggestions to tone down the implied criticism and to highlight the differences in saturation values.

Changes to the manuscript: We have revised the paragraph discussing the shape results as suggested.

Referee Comment 9: Ratio of timescales: The authors propose a framework in which the ratio between regolith residence time and particle detachment time determine initial sediment sizes. I think this framework is fine; however, the way it is presented assumes that regolith is necessary for weathering. I am still unconvinced that fracture spacing does not include effects from mechanical weathering (frost cracking, tree roots, thermal fluctuations, etc.). Perhaps the authors just need to clarify when the particle size clock starts (see next suggestion below). If it starts after particles are released from bedrock, then their framework makes sense. However if it starts when pure bedrock first begins to weather/crack, it may not be appropriate.

Author answer: As discussed above, we need to more clearly define how we conceptualize the potential role of some (but not all) weathering processes in shifting the initial sediment size distribution away from the size distribution of latent, fracture-bound blocks in bedrock. For the time scale of particle production, Tp, the "particle size clock" starts, during exhumation of the rock, when the boundary between intact rock (weathered or unweathered) and mobile regolith (whether remaining in contact with the bedrock or immediately removed by active transport) reaches outermost surface of the particle to be detached. This is implicit in the definition of Tp as the time required to detach a layer of particles of a given size. This is distinct from the two possibilities suggested by the Referee (after detachment or at a depth below any influence of weathering).

Changes to the manuscript: As noted previously, in the revised introduction we explain more carefully and completely our conceptualization of the potential role of weathering in altering the initial size distribution away from the latent size distribution of fracturebound blocks, as distinct from the potential role of weathering processes in contributing to the fractures bounding latent sediment particles. We have also made changes in the discussion to clarify the definition of the particle production time scale.

Referee Comment 10: Definition of "latent" and "initial": It's currently a bit difficult to understand what the authors mean by "latent" and "initial." A clear definition in the introduction would help a lot.

Author Answer: See our response to comment 1 above.

Changes to the manuscript: In the revised introduction we provide clear definitions of these two terms.

Referee Comment 11: Talus sampling: In line 139 the authors explain that spatially uniform sediment sampling along the talus slope should yield an accurate grain size distribution even with size selective entrainment. However, this assumes that size distributions change linearly downslope. The authors might want to simply point out this assumption.

Author Answer: A linear variation is not required for spatially uniform sampling to accurately characterize an attribute of a single population distributed non-uniformly in space. Consider the problem of sampling a stream bed to determine the median particle size representative of the entire bed. Lateral sorting processes, active throughout the channel, may distribute the different particles sizes across the bed in a highly non-linear pattern of patches and gradients. The most straight-forward sampling approach in this case is a uniform grid, as detailed by Bunte and Abt (2001) in Chapter 6 of their comprehensive sediment sampling manual.

Changes to the manuscript: We have edited this passage to clarify this point, and have added a reference to Bunte and Abt (2001).

Referee Comment 12: Figure 3: The authors refer to parts of the figure by letter, but they're not included in the figure.

Author answer: Thank you for catching this omission.

Changes to the manuscript: We have added letter labels to each panel in the revised version of Figure 3.

Referee Comment 13: Figure 4: A legend that identifies data points is needed. It is unnecessarily difficult to refer to figure 1 and remember the colors and shapes to understand figure 4.

Author Answer: We agree, a legend for figure 4 is helpful.

Changes to the manuscript: We have added a legend to the revised version of Figure 4.

Referee Comment 14: Missing references: The authors should also cite some missing recent relevant studies: Shobe et al., 2016, who explore hillslope sediment size controls on river incision; Glade et al., 2017, who show data for boulder size distributions in an exposed bedrock system; and Glade and Anderson, 2018, who discuss the implications of weathering vs. erosion rate timescales on hillslopes; Ward 2019, who discusses ratios between incision rate and cliff retreat timescales; Duszynski et al., 2019 who review scarp retreat mechanisms and the role of weathering.

Suggested references: Duszynski, Filip, Piotr Migon, and Mateusz C. Strzelecki. "Escarpment retreat in sedimentary tablelands and cuesta landscapes–Landforms, mechanisms and patterns." Earth-Science Reviews 196 (2019): 102890.

Glade, R. C., and R. S. Anderson. "Quasi-steady evolution of hillslopes in layered landscapes: An analytic approach." Journal of Geophysical Research: Earth Surface 123.1 (2018): 26-45.

Glade, Rachel C., Robert S. Anderson, and Gregory E. Tucker. "Block-controlled hillslope form and persistence of topography in rocky landscapes." Geology 45.4 (2017): 311-314.

Shobe, Charles M., Gregory E. Tucker, and Robert S. Anderson. "Hillslope-derived blocks retard river incision." Geophysical Research Letters 43.10 (2016): 5070-5078.

Ward, Dylan J. "Dip, layer spacing, and incision rate controls on the formation of strike valleys, cuestas, and cliffbands in heterogeneous stratigraphy." Lithosphere 11.5 (2019): 697-707. Interactive comment on Earth Surf. Dynam. Discuss., https://doi.org/10.5194/esurf-2020-54, 2020. C6

Author Answer: Thank you for suggesting these additional references.

Changes to the manuscript: In the revised introduction we now cite Shobe et al. (2016) and Glade et al. (2017) as illustrations of the importance of initial sediment size to the geomorphic evolution of hillslopes and rivers, and in the discussion of weathering and sediment production time scales we cite the work of Glade and Anderson (2018).

---

## Author Comment (AC2) · 4 Jan 2021

Verdian et al. Anonymous Referee #2

Referee Comment (overview) Verdian et al., present a timely new dataset that measures bedrock fracture spacing on cliffs and compares the spacing of fractures to the

grain size of sediment contributed to downstream hillslopes. The topic is very pertinent. Growing research shows that sediment grain size can impact relationships between topography and sediment transport at a range of spatial scales, and the need is growing for empirical datasets that constrain the initial grain size of sediment produced from fresh rock. This study primarily tackles the 2nd point above by collecting field data from a number of cliff locations previously studied by Moore et al., 2009. I found the connection between bedrock fracture spacing and talus-sediment grain size convincing and promising, but have suggestions to hopefully strengthen the manuscript's clarity, impact, and context.

After some moderate revisions that address these comments, I would be happy to work with these authors towards publishing this manuscript in esurf.

Review aspects: Does the paper address relevant scientific questions within the scope of ESurf? Yes. Does the paper present novel concepts, ideas, tools, or data? Yes. New data. Are substantial conclusions reached? Yes. Are the scientific methods and assumptions valid and clearly outlined? Could use clarification Are the results sufficient to support the interpretations and conclusions? Yes Is the description of experiments and calculations sufficiently complete and precise to allow their reproduction by fellow scientists (traceability of results)? Generally, yes. Do the authors give proper credit to related work and clearly indicate their own new/original contribution? Yes. Does the title clearly reflect the contents of the paper? Yes. Does the abstract provide a concise and complete summary? Generally, yes. Is the overall presentation well-structured and clear? Yes with suggestions Is the language fluent and precise? Yes but could improve Are mathematical formulae, symbols, abbreviations, and units correctly defined and used? Yes Should any parts of the paper (text, formulae, figures, tables) be clarified, reduced, combined, or eliminated? No Are the number and quality of references appropriate? Yes with some additions Is the amount and quality of supplementary material appropriate? Yes

Principle criteria Scientific significance: good – ideas and concepts have existed and

have been explored in other publications, but this study contributes useful new field datasets – particularly fracture spacing measurements paired with talus grain shape and size measurements. New conceptual model introduced in discussion section requires some clarification, connection to hillslope/catchment-scale variability in weathering zone thickness, and specific factors that can be measured in the field.

Scientific quality: good – individual data collection methods are valid, but comparison between datasets and connection to pre-existing work should be clarified. Clarify quantitative methods used to distinguish results and make comparison between datasets. Datasets are very useful, but additional context is likely required to communicate results and interpretation to readers. I would like to see some more detail in the methods section.

Presentation quality: good/fair – Clear statement of testable hypothesis. I recommend adding to the background/study site section to frame tectonic, climatic, and lithologic context. Some organizational suggestions to distinguish results from interpretations (results section vs discussion section). Encouraged to add subheadings to discussion section and results section to improve organization of results and interpretations. Some additional context is needed to communicate the importance of the problem and the utility of the findings.

Author Answer: We are grateful to Referee #2 for these thoughtful, thorough, and constructive comments. They are very helpful for improving the clarity and rigor of the manuscript. We have considered every comment carefully, and made changes to address each suggestion and concern as appropriate. In responding to many of the comments below, we make reference to our responses to comments by Referee #1; we do not repeat all of those responses in full here for the sake of brevity.

Line by line edits:

Abstract

Referee Comment 1: Line 19: Before introducing results (median fracture spacing and particle size. . .), add a sentence describing methods used to measure fracture spacing (scan lines) and sediment grain size (a-b-c axis measurements). Potentially this could be incorporated into the previous sentence.

Author Answer: Good suggestion.

Changes to the manuscript: We revised the abstract to include more detail on methods.

Referee Comment 2: Line 19: Here and throughout the manuscript, consider switching "particle" with grain or clast? I would interpret particle to include wood or other debris that does not originate on cliffs. Just my preference though.

Author Answer: We did consider this suggestion. Clast is often used to refer to particles lithified within a sedimentary rock, just as grain is often used to refer to individual crystals within a crystalline rock. We prefer the more generic particle, and make clear that we are referring to sediment particles which are fragments of rock.

Changes to the manuscript: We did not make changes in response to this comment.

Referee Comment 3: Line 20: Is there a metric you can provide to quantitatively describe the correspondence between fracture spacing and b-axis diameter?

Author Answer: If there were more words allowed in the abstract, we would include more of the detailed quantitative results. However, the finding that median b-axis is systematically larger than median fracture spacing is an important result.

Changes to the manuscript: We have revised the abstract to include this result.

Referee Comment 4: Line 22: "weathering has not modified latent sediment either before" seems confusing to me. You are measuring bedrock fracture spacing on the cliff faces (which are affected by near-surface weathering to some degree), so you haven't quantified how the latent sediment size changes before reaching the cliff face. It is important to define (possibly in the abstract but definitely in the introduction) at

what point in rock exhumation to the surface you're measuring latent sediment size (see later comment)

Author Answer: We agree that our original presentation of the notion of a latent sediment size was confusing.

Changes to the manuscript: As detailed in the responses to Referee #1, we have made substantial revisions to address this issue, to the introduction as well as the abstract.

Referee Comment 5: Line 22: I would remove point about modification of sediment before detachment and focus on point about modification of sediment during detachment: i.e. grain size does not fine significantly during rockfall or clast spallation from the cliff face

Author Answer: Agreed.

Changes to the manuscript. We have revised the abstract as suggested.

Referee Comment 6: Line 24: replace "it" with clasts? - "clasts contain some fractures inherited from bedrock" – is there any field evidence of this and is it possible to present this evidence (even if photographs of deposits)

Author answer: There is field evidence, as noted in our responses to Referee #1.

Changes to the manuscript: We have added a field photo to figure 2 showing an example.

Referee Comment 7: Line 27-30: Possibly you can shorten these sentences and allow space to expand on your methods earlier in the abstract. Also, it's important to acknowledge landscapes between the two end-members mentioned, as many steep catchments fall into this domain space (some cliffs, but still some soil-mantled hillslopes).

Author Answer: Agreed.

Changes to the manuscript. We have revised the abstract as suggested.

Introduction

Referee Comment (Set) 8: Line 33: I might reword "the life of sediment" and remove this opening phrase: (The size distribution of sediment influences chemical, physical, and . . . ect) Line 36: Replace "The first stage begins" with (Initial sediment grain size is set when clasts are detached from. . . ect..) Line 38: rephrase?: "The resulting initial size distribution is the starting point for the evolution of the size distribution of sediment on hillslopes, and therefore sets the upper limits on sediment size distribution as sediment is routed through the catchment network" ? – slightly more specific than the life-cycle of sediment Line 41: good statement of overall problem that this paper is addressing Line 46: Can you cite this point? Or add a qualifier to the beginning of this sentence: "We hypothesize that. . ." or "The "latent" size distribution of clasts is thought to be set by . . . (add a citation to the Sklar et al., 2017 paper maybe if no others come to mind?) Line 48: I feel like this paragraph should likely be split into at least 2 if not 3 or 4 paragraphs. Here might be a place to do it. Some place in this paragraph you need to specify where in the rock exhumation timeline you are considering fractured rock to reflect a latent clast size.

Author Answer: These are all thoughtful suggested edits.

Changes to the manuscript: We have revised the introduction substantially, and taken these suggestions into account.

Referee Comment 9: It probably makes sense to define the time to measure "latent" sediment size at the point when "clasts" are at the cliff face (this is where you are measuring fracture spacing), but it's important to note that near-surface processes that fracture bedrock can modify fracture spacing before rock reaches the cliff face surface where your fracture spacing measurements are occurring. The true "latent" clast size reflecting the first interaction between the topographic surface and fractures likely occurs at some depth beneath the cliff face, where stress induced by non-flat topography becomes non-negligible (i.e. Miller and Dunne, 1996 – https://doi.org/10.1029/96JB02531) or temperature fluctuations can fracture rock due to frost cracking (Hales and Roering, 2007 – https://doi.org/10.1029/2006JF000616 ) or diurnal heating and cooling (harder to get specific depths below surface where this process is important?. . . Collins and Stock, 2016 – rockfall triggered by thermal cycling, or work by Epps: Epps and Keanini, 2017?).

I think it is good to pick cliff surfaces as the surface where "latent" clast sizes are quantified (because this is most possible to measure), but it's important to clarify that modification of clast sizes related to near-surface processes may have potentially started at deeper depths, and the amount of subsurface latent clast size modification before reaching a cliff face might depend on climate, hillslope relief, or tectonic stress orientations.

Author Answer: These same issues were raised by Referee #1. Please see our reply to their comment 3.

Changes to the manuscript: We have revised the introduction to clarify our definition of latent size accordingly.

Referee Comment (Set) 10: Line 48: If you describe the point above at the end of the 3rd sentence in the current paragraph, I think it makes sense to split the next sentence into a new paragraph (describing processes that fracture rock unrelated to the near surface – tectonic fracturing, unloading, and cooling) Line 54: I would split the sentence starting with "As weathering commences" into a new paragraph that talks about near-surface processes that fracture bedrock. (climate, topographic stress perturbations, deep weathering) Line 60: New paragraph starting with "Although these hypotheses.." End this new paragraph re-iterating that you are defining the "latent" sediment size as the spacing of fractures on exposed cliff faces?

Author Answer: These are all reasonable ideas for copy-editing the original manuscript.

Changes to the manuscript: We have taken these suggestions into account in the revision of the introduction.

Referee Comment (Set) 11: Line 61-62: I believe this point has been explored by Messenzehl et al., 2018 and Neely and DiBiase, 2020 (this is in preprint.. so I'm not sure if you can cite this contribution yet?, but I see this is cited elsewhere due to relevance). I would reword to say: "the relative importance of latent sizes and weathering in initial size distributions has been explored systematically across a limited suite of climatic and lithologic settings" ? Line 66-67: maybe re-iterate at the end of this sentence that prior studies focused on a limited range of bedrock lithology and climate variables?

Author Answer: We agree with the goal of these suggestions of describing previous work accurately. We did not mean to imply that previous work was not systematic.

Changes to the manuscript In the revised introduction, we now contrast our study with the limited range of lithologies and weathering environments studied in previous work.

Referee Comment 12: Line 70: It might be important to note the resolution of the fracture spacing surveys in this study (could only resolve fractures with apertures > xxx cm). Your fracture spacing measurements occur at much higher resolution.

Author Answer: This comment regarding Sklar et al. (2020) is only true of the fractures measured from air photos. That paper also reports fractures measured in situ with similar resolution to those reported here. (Note that we did not measure fracture opening width or aperture, rather we measured the length of rock between successive fractures, many of which were closed, with submillimeter aperture.)

Changes to the manuscript: We did not make changes in response to this comment.

Referee Comment 13: Line 73: Same comment as above (could only resolve fractures with apertures > 1-3 cm).

Author Answer: The issue of resolution isn't essential to this introductory review of the results of previous work. It is more relevant in the methods section.

Changes to the manuscript: We have revised the relevant passage in the methods section to also include note of the resolution of fracture detection in previous work.

Referee Comment 14: Line 73: change DiBiasi to DiBiase

Author Answer: Thank you for catching this misspelling.

Changes to the manuscript. DiBiase is now spelled correctly throughout.

Referee Comment 15: Line 74: Results from Neely and DiBiase show that latent clast sizes dominate on steep slopes where bedrock cliffs are exposed, but not on gentle soil-mantled slopes.

Author Answer: Agreed.

Changes to the manuscript. Sentence revised as suggested.

Referee Comment 16: Line 74: May add reference and background to Attal et al., 2015 (https://doi.org/10.5194/esurf-3-201-2015 ) and Roda-Boluda et al., 2018 (https://doi.org/10.1002/esp.4281 ), which look at erosion rate and hillslope morphology controls on sediment grain size. These studies do not quantify bedrock fracture spacing, but their results imply a link between weathering on soil mantled slopes and sediment grain size.

Author answer: Both of these references are relevant to the discussion where we consider the role of erosion rate in regulating residence time and thus the degree of influence of weathering.

Changes to the manuscript: In the revised manuscript we cite both of these previous studies in the discussion section.

Referee Comment 17: Line 76: I like contrasting the Messenzehl findings with the prior findings from California landscapes. Maybe move the following sentence (lines 78-79) up to the end of this paragraph to state how your study fits in with these prior investigations?

Author Answer: Thank you for this suggested edit.

Changes to the manuscript: We have revised this section of the introduction, as detailed in responses to Referee #1.

Referee Comment 18: Line 81-82: change "initial sizes" and "latent sizes" to "talus clast sizes" and "latent clast sizes" ?

Author Answer: As noted above, we prefer not to use the term "clast" in this way.

Changes to the manuscript: We have not made changes in response to this comment.

Referee Comment 19: Line 85: Also, would mention that clast detachment could occur along pre-existing fractures that are below your detection limit. The small-aperture fractures may also be pre-existing.

Author Answer: This is more of a methodological issue. These alternative hypotheses are not method dependent.

Changes to the manuscript: We have not made changes in response to this comment.

Referee Comment 20: Line 87: Another reason talus grain size may not match fracture spacing is if talus sediment is sorted after detachment. This is mentioned later in the manuscript, but should be mentioned here as well.

Author Answer: Agreed.

Changes to the manuscript: Passage revised as suggested.

Referee Comment 17: Line 87: new paragraph at "Neither of these alternatives"?

Author Answer: Agreed.

Changes to the manuscript: New paragraph begun as suggested.

Methods

Referee Comment (Set)18: Line 98: Before the methods section, I think readers need

a "study site" section that describes the various tectonic, climatic, and lithologic settings of each site. (Analyzing cliff/talus systems across these different variables is a main strength of the paper!). This section should at a minimum: - Detail the location and tectonic setting/history of the outcrops (how does this connect to the inherited bedrock fracture network? (maybe add a map figure to show this too).. are some outcrops closer to active or inactive (– dead) faults? - Details the climate and climate history of outcrops – relation to frost-cracking? diurnal heating and cooling? Some of this information is in table 1, but the climate history may also be important. - Introduces the prior work used from Moore et al., 2009 (how did Moore et al., 2009 estimate cliff retreat rates used in later parts of the manuscript?) Line 103: Include a source after the statement that says three sites where differences in lithology correspond to differences in average fracture spacing?

Author Answer: Agreed, these are helpful suggestions.

Changes to the manuscript We have added a new sub-section at the beginning of the methods section on study sites that describes the tectonic, lithologic and climatic settings of each site, with relevant citations.

Referee Comment 19 Lines 108 – 120: Scan lines have been used to measure bedrock fracture spacing in a number of applications, and it would probably be good to cite some of the studies that developed/used these methodologies.

Author answer: Agreed.

Changes to the manuscript. We have added citations to two additional prior studies that used the scan-line technique to measure bedrock fracture spacing

Referee Comment (Set) 20 Lines 108-120: One of my main critiques of this study is that the comparison is not straightforward between sediment grain size along an A-B-C axis and fracture spacing on a scanline. This needs to be stated clearly, because this impacts how to interpret results presented from the study. Scanlines the way they are

described (to my knowledge in this manuscript) do not usually run through the longest or shortest axis of a fracture-bound block on a cliff. A horizontal scanline will likely be skew across the fracture-bound block unless all fractures are perfectly vertical. The spacing between the fractures on the horizontal scanline is usually not the A-B or C axis of the fracture-bound block. See Figure 2 in the manuscript, lower right. While the scanline fracture spacing is still a useful measure of fracture spacing, I'm not surprised that the B-axis of the clasts is larger than the scanline-fracture spacing in the results, because the scanline-fracture spacing likely does not sample the B-axis of the clast (scanline crosses corners of clasts and does not go along the widest or shortest axis). For example, the Neely and DiBiase study in preprint measured fracture spacing as the short-axis of fracture-bound blocks on orthophotos of cliff faces. This assumes that the short-axis of fracture-bound blocks on the cliffs represents the B-axis. Though this isn't true for "latent" particles where the visible short-axis of the fracture-bound block is actually the C-axis, measurement of the short axis of the fracture-bound block might be more straightforward to map to the B-axis of a detached clast (see figure attached). The difference between these methods needs to be clarified for the reader here, and this should be mentioned when interpreting the results (unless I'm missing something about how fracture spacing was measured)? (See attached figure) If scaled photographs exist for part or all of the scanlines (i.e. figure 2 lower left), another option would be to measure the short and long axis of clasts along the scanline on the photographs (with a photo resolution limit of 2mm) (i.e. attached figure right panel). This could be compared to the scanline fracture spacing and the A-B-C axis of the clasts in the talus piles. This could be a useful comparison for future studies connecting fracture spacing to geomorphic processes or sediment grain size.

Author Answer: We agree that the scanline method is an imprecise approach to characterizing the dimensions of any given fracture-block exposed on a cliff face. Analysis of scaled photos, as done by Neely and DiBiase (2020), permits more precise measurements, at least for large blocks, although there remains the question of which axes are being measured. And, as we discuss in our response to comment 6 by Referee

**1, non-orthogonal joint sets may create fracture-bound blocks that do not present any of the three primary axes to the viewer. "In the most general case, blocks are formed by three intersecting joint sets with non-perpendicular orientations, and thus are not rectangular prisms. In this case, the inter-fracture distance measured along any given scan line crossing a block could possibly range from near zero (near the tip of an acute-angled point) to greater than the a-axis length (for example if spanning the longest possible linear distance across a rectangular block face). An essential assumption in using the scan-line technique is that this variability can be overcome by a large sample size, resulting in an accurate if imprecise estimate of the central tendency and spread in the underlying population of fracture spacings. However, biased estimates can result at any single site, for the reasons outlined above." Additional analysis using scaled photographs is beyond the scope of this paper. However, we are confident that our large sample sizes are sufficient to test the null hypothesis that the distributions of fracture spacing and talus particle size are correlated across our sites.**

Changes to the manuscript In both the methods and discussion sections, we have added text to clarify the differences between in-situ scanline and scaled photo methods and the potential sources of bias in comparisons of scan-line derived fracture spacings and talux b-axis diameters.

Referee Comment 21: Line 124: Slope-parallel transects result in 300 clasts measured per talus pile? Correct? Maybe state this if true?

Author Answer: Our method resulted in measurements of 100 talus particles per talus pile. Evidently, our original text describing the method (sum the three tape lengths and divide by 100 to obtain the spacing between samples) did not make this outcome clear.

Changes to the manuscript: We have revised this passage to state explicitly that, for each cliff-talus pair, 100 talus particles were measured in total, summed across the three transects.

Referee Comment 22: Line 124: again I'm preferential to saying clasts instead of particles (because particles may include wood or really anything that doesn't necessarily come from the cliffs)

Author Answer: We appreciate the referee's preference for the term clast, but are not concerned that readers will think we're measuring pieces of wood if we continue to use the term particle.

Changes to the manuscript: We have not made any changes in response to this comment.

Referee Comment 23: Line 128: "everything else" – I'd replace this with "larger clasts"

Author Answer: Agreed, "everything else" is imprecise.

Changes to the manuscript: Sentence changed to "larger particles"

Referee Comment 24: Line 129: to be clear, "particles" are "talus cone clasts"? Also I'm confused, if the fracture spacing resolution limit was 2 mm, how do you compare talus clasts <2 mm to the "latent" clast size, which cuts off at a detection limit of 2 mm (presumably) ?

Author Answer: The point here is that there is no benefit to measuring particle diameters less than 2mm because there are no fracture spacings to compare to. The comparison of the <2mm fine tail of the distributions thus involves a possibly non-zero fraction of the talus size distribution versus zero in the fracture spacing distribution.

Changes to the manuscript: This sentence has been revised for clarity.

Referee Comment (Set) 25: Lines 134-139: I agree with this section, but how do you know that the talus cone captures all of the sediment grain size distribution spalled from the cliff? . . . the largest clasts roll the furthest, do some of these traverse the whole talus pile? If the talus pile is mined by an active stream at its base, the largest clasts may be somewhere else downstream and the talus pile undersamples the coarsest clasts. It's been shown in a couple studies/settings that the coarsest

sediment grain size distribution is typically found at the base of colluvial/headwater channels in steep landscapes: Hack 1965 https://pubs.usgs.gov/pp/0484/report.pdf), Brummer and Montgomery, 2003, https://doi.org/10.1029/2003WR001981 , and Neely and DiBiase preprint figure 9 https://www.essoar.org/pdfjs/10.1002/essoar.10502617.2 It might be important to note this limitation when comparing bedrock fracture spacing to the grain size of sediment on talus cones. But maybe these talus piles act as better 'clast traps' than angle of repose talus cones... Also, it looks like the bottom two sites in figure 1 have roads at the bottom of them? I'm not sure if large clasts can travel far enough from the cliffs to reach these roads, but these clasts would likely be cleared by road crews?

Author Answer: At the Sierra Nevada sites, and the Grizzly Peak site, there are no streams or other sediment sinks at the base of the talus slopes. These talus slopes have clearly defined downslope end points to the depositional feature. This is does not mean that some especially large and energetic particles may have traveled past the end of the slope and come to rest far beyond the end of our measurement transect. Likewise, at the Twin Peaks sites, where there is a road adjacent to the toes of the talus slopes, some large particles may have been lost from the deposit. In these cases, it's possible that we have under-sampled the coarse tail of the distribution. This would not affect the estimate of the median size, and would likely have minimal impact on the distribution spread.

Changes to the manuscript We have included mention of this issue in the expanded site description sub-section.

Referee Comment 26: Line 140-145: These methods are mentioned, but I don't see the results presented anywhere. Or the methods detailing how these results are integrated into the full distributions are missing? Does this particle size distribution replace the fine tail of the talus grain size distribution that is below the resolution limit of the fractures (2 mm?). Do you quantify fracture spacings finer than 2 mm? even though this is the detection limit for the fractures?

Author Answer: The outcome of this sampling was reported in the discussion section of the original manuscript where we discuss possible explanations for the coarse offset of the median b-axis diameters from the median fracture spacing. We were not able to obtain subsurface samples at every site so we did not combine those measurements with the surface point counts. However, we used them as a possible indicator of vertical sorting which might explain the b-axis offset. As described on line 255 of the original manuscript, the surface and subsurface distributions overlap sufficiently and thus do not indicate a truncation of the fine tail of the surface talus size distribution due to vertical sorting processes. Because our analysis of the subsurface samples is limited to addressing this question, we do not use those data to make any comparisons with the fracture spacing distribution.

Changes to the Manuscript: We have rewritten the relevant passages to provide a clearer explanation of this analysis.

Referee Comment 27 Lines 147 – 151: This section seems out of place. Maybe move this to be near the other paragraphs that describe how you measure sediment grain size (1st paragraph in this section?)

Author Answer: The first three paragraphs of this section all concern the methods employed at the Sierra Nevada sites. The last paragraph describes methods employed at the Bay Area sites.

Changes to the Manuscript: We have edited the text to make it clear that the first three paragraphs all apply specifically to the Sierra Nevada sites.

Referee Comment (Set) 28: Methods section suggestions overall: - Need to clarify difference between scanline fracture spacing and a-b-c axis measurement of clast/fracture-bound block - A number of statistical techniques are later mentioned in the results section. It would help me to introduce these techniques in the methods section and explain how these techniques are used to quantitatively distinguish between the different distributions and test the hypothesis of the paper? - Detail how the "fines"

are accounted for. Where are the bulk sample sediment results presented? How are these included into the distribution? Is the same thing performed with the fracture spacings? - Note that talus cones may not capture the largest grains if the base is actively mined by a steep and competent stream (or a road maybe?.. not sure)

Author Answer: Thank you for this concise summary. We have addressed these three of these four points in our responses to comment numbers 20, 25, and 26. We address the question of whether to include information about how we analyzed the data in the methods, or keep it in the results, in our responses to comments 30 and 31 below.

Changes to the Manuscript: See responses to comments 20, 25, 26, 30 and 31.

Results

Referee Comment 29: Line 155: "c-axis diameters as small as 2 mm" – what about the bulk sediment samples (fines?)

Author Answer: See response to comment 26

Changes to the Manuscript: None.

Referee Comment 30: Line 161: cumulative empirical distribution function is not a jargon-y term? May be helpful to define in the methods section?

Author Answer: We could have referred to it as simply as cumulative distribution, or used the acronyms CDF or EDF but those would be imprecise. We chose to use the complete and correct name to avoid any confusion. Readers unfamiliar with this and other standard statistical quantities can easily look them up (e.g. https://en.wikipedia.org/wiki/Empirical_distribution_function).

Changes to the Manuscript: We did not make any changes in response to this comment.

Referee Comment 31: Lines 159-168: A lot of this comparison is qualitative. I see more quantitative comparison in the next paragraph, but I feel unprepared to understand

this comparison because the methods section did not introduce how the quantitative comparison would occur.

Author Answer: A common convention in much scientific literature is to use the methods section to describe how the data are obtained, and to report how the data are analyzed in the results section. We prefer to use this approach, rather than introduce the data analysis techniques in a generic way in the methods, without the data in hand to understand why those techniques might be used. In addition, data analysis choices often flow from the results of preceding analyses, which cannot be explained in a methods section before the reader has seen those preceding results. The challenge of course, is to explain the data analysis clearly and this comment, along with others by Referee #1 indicate that the original manuscript did not.

Changes to the Manuscript: As described in our response to Referee #1, comment 6, we have substantially expanded the explanations of the choices and techniques used in analyzing the data.

Referee Comment 32 Line 167: I would like to see the discussion section return to the anomalous result from MT-39 – do the clasts here have many fractures still retained ? – if the fracture spacing is smaller than the c-axis even?

Author Answer: Excellent suggestion.

Changes to the Manuscript: We have added a photo from MT39 to Figure 2 which shows fractures evident in the large boulders on this talus slope.

Referee Comment 33 Line 173: I'm not quite sure how these "p" values were calculated or what they mean. I see some comparison to a 1:1 correlation and I think I know how you could do this, but in the methods section, could this be clarified and related to how you are testing your hypothesis?

Author Answer: Please see our response to Referee #1, comment 6, which addresses this comment.

Changes to the Manuscript: Please see our response to Referee #1, comment 6, which addresses this comment.

Referee Comment 34 Line 178 – 181: this is an interpretation and could be moved into a discussion section? – I am a fan of splitting the results and discussion sections.

Author Answer: We are fans of telling an engaging story, which often means providing interpretations of some results in order to motivate additional analyses that lead to additional results. A 'just-the-facts' approach to reporting results can make more work for the reader, who might have to hold in mind many disparate observations while taking in others, until at a later point in the paper the authors view of their meaning is revealed. Identifying how specific results answer specific questions as the story unfolds can then make it possible to focus in the discussion section on the larger context and implications. To indicate that we have chosen this approach, we titled this section "results and interpretations." For a compelling argument for this story-telling mindset in writing science papers, we enthusiastically recommend the book "Writing Science" by Joshua Schimel.

Changes to the Manuscript: We did not make any changes in response to this comment

Referee Comment 35: Line 185-186: Would be helpful to clarify how p-values are calculated in methods section

Author Answer: As noted above and in our response to Referee #1, comment 6, we prefer to explain the data analysis where the results are analyzed. We appreciate that the hypothesis tests were not explained thoroughly.

Changes to the Manuscript: We have expanded the explanation of this and other hypothesis tests.

Referee Comment 36: Lines 193-194: Would be helpful to clarify in the methods section how the Weibull distribution is populated/used to compare with your data? I find this hard to interpret without some background on the methods.

Author Answer: We agree that readers unfamiliar with the Weibull distribution would not be well served by the text in the original version of the manuscript.

Changes to the Manuscript: We have revised this section to explain more carefully how we compared our data to the theoretical distributions, and how to interpret the Weibull plotting space; we also provide reference for readers who want more background.

Referee Comment 37: Line 194-195: Change "the degree to which the data follow a Weibull distribution at each site is illustrated in Figure 5" to a figure call to figure 5 at the end of the previous sentence "(Fig. 5)"

Author Answer: We prefer to direct the reader to Figure 5 for the point of illustrating the quality of the fits. The previous sentence reports the results of the analysis that used a statistics software package, not the graphical analysis of Figure 5.

Changes to the Manuscript: We did not make any changes in response to this comment.

Referee Comment 38: Lines 190-210: Be clear to specify that "A,B,C-axis" distributions come from talus sediment and not from fracture-bound blocks on cliffs. It's not clear to me how the Weibull distributions are quantitatively linked to the results presented? Most of the comparison seems qualitative at this point? Is there a way to quantify this comparison like with the p-values? If so this should be reported and introduced in the methods section.

Author Answer: We appreciate that this section of the original text was unclear. The quantitative link between the theoretical Weibull distribution and the empirical data lies in the degree of linearity of the data when plotted in this space. The purpose of Figure 5 is to allow readers to evaluate for themselves the how the data compare to pure linear trends. We focus on this qualitative approach to evaluating the data because hypothesis tests of whether the data match the theoretical distribution are not that informative. Ironically, a larger data set is more likely to fail the test (i.e. to result in a

rejection of the null hypothesis that the data are indistinguishable from the theoretical distribution) because the standard of proof becomes so high that even slight deviations cause the hypothesis to be rejected.

Changes to the Manuscript: We have substantially revised the text addressing the comparisons of the data with theoretical distributions, and have added more detail regarding the quantitative tests that lead us to use the Weibull plotting space to evaluate the empirical distributions.

Referee Comment 40: Lines 200 and 202: "In some cases" – is there anything specific about these cases? Should these be discussed more in the discussion section?

Author Answer: Our point here is to help the reader to understand how the patterns of data plotted in the Weibull space link to previous results, such as the differences in the spreads of the distributions documented in Figure 3 and Figure 4, panel d.

Changes to the Manuscript: This passage has been revised for clarity.

Referee Comment 41: Lines 217: reword? "Within each rock type, there is little site-to-site variability in mean particle shape. When sites are grouped by rock type, we find . . .."

Author Answer: Agreed.

Changes to the Manuscript: Sentence changed as suggested.

Referee Comment 42: Line 220: how much lower are the mean b:a, c:a ratios?

Author Answer: For b:a ratio, metasediment = 0.57 and granodiorite = 0.64 For c:a ratio, metasediment = 0.26 and granodiorite = 0.35

Changes to the Manuscript: These values have been added to the text in parentheses.

Referee Comment (Set) 43: Line 221-223: this is an interpretation and could be moved to a discussion section?

Lines 229 – 234: this point also reads like a discussion point (contextualizing with prior work) – reorganize?

Author Answer: As in our response to comment 34, we prefer to link narrow results with their specific interpretations when the discussion focuses on the larger context and bigger picture implications.

Changes to the Manuscript: We did not make any changes in response to this comment.

Discussion:

Referee Comment 44: Overall – I had a hard time following parts of the discussion. Potentially, points should be organized into subheadings? - Significance/reliability of comparison between scan line fracture spacing and sediment grain size in talus piles? - Comparison with observations from other landscapes where bedrock fracture spacing was quantified? - Reasons for differences between fracture spacing and sediment grain size? (sorting, kinetic sieving, incomplete breaking along fracture planes, measurement resolution?) - Climate/weathering controls on initial sediment grain size? - Conceptual model (with some revisions/clarification?)

Author Answer: We agree that subheadings would improve the clarity of the discussion section.

Changes to the Manuscript: We have added sub-headings to discussion

Referee Comment 45: Line 237: "nearly the full network of fractures" .. Rephrase: "from the network of fractures with apertures >2 mm"?

Author Answer: The detection limit of 2mm refers to the minimum length along the scan line of solid rock between successive fractures that we counted as a spacing between fractures. This does not refer to the width of the fracture opening, or aperture.

Changes to the Manuscript: In the methods section we have clarified this detail.

Referee Comment 46: Lines 237-238: "This finding" and "it" are hard to unpack in this sentence. I'm having a hard time interpreting this sentence.

Author Answer: Agreed.

Changes to the Manuscript: The sentence has been revised for clarity

Referee Comment 47: Lines 241 – 247: This might be a good place to return to the discussion about the difference between scan-line fracture spacing and the long and short axis of a fracturebound block. Note that Neely and DiBiase (check spelling) uses the short axis of fracture-bound blocks, which may account for the difference between these study results. Line 249: See above comment, also Neely and DiBiase –preprint - and Sklar et al., 2020 also have coarser fracture detection limits? (1-3cm in Neely and DiBiase (typically 2 cm) and not sure about Sklar et al., 2020 because a variety of techniques are used – aerial imagery likely has a coarser detection limit though?). The fracture spacing will increase with coarser fracture detection limits – but maybe fracture scaling relationships could be used to compare between these studies – probably a non-trivial endeavor (see Hooker et al., 2014 and similar work that quantifies the distribution of fracture apertures in rock masses https://doi.org/10.1130/B30945.1 ) Either way, I would make sure to acknowledge the difference in technique used to measure fracture spacing and the difference in minimum resolvable fracture aperture when comparing these results to results from these other two studies.

Author Answer: Agreed.

Changes to the Manuscript: This section has been expanded to consider alternative explanations for the offset between median talus b-axis and median fracture spacing, including the comparison between scan-line and scaled photo measurement techniques.

Referee Comment 48: Lines 251-252: Not sure where these results are presented??

Author Answer: As noted in our response to comment 26, this is where the outcome of the subsurface particle size analysis is reported.

Changes to the Manuscript: No changes were made in response to this comment.

Referee Comment 49: Line 257: are there field photographs of this? These would be useful to show if even qualitative?

Author Answer: Agreed.

Changes to the Manuscript: As noted elsewhere, we have added a field photo to Figure 2

Referee Comment 50: Lines 261-265: this reads more like a results section. I think it would be useful to incorporate the climate data comparison with fracture spacing/ talus grain size in some sort of figure? Plot mean annual temp & precip. Vs median fracture spacing/talus B-axis ? Same thing with lithology vs median fracture spacing/talus B-axis? This would show any correlation or lack of correlation between these variables and fining the sediment after it spalls off of the cliffs? Again, I think the study sites ranging across various climates and lithologies is a strength of this paper and should be used/analyzed! – though some more analysis might need to be done to correctly synthesize these additional parameters/forcings.

Author Answer: This study was not designed to test for the influence of climate on fracture spacing or talus particle size. We would expect climate to play a role if weathering were a significant factor, but by selecting bare bedrock cliff faces, known to be actively eroding, we sought to minimize the potential for weathering to be a significant factor. The fact that the climate variables do not correlate with the regression residuals from Figure 4 confirms that assumption, but does not shed any light on the question of how weathering would influence initial particle sizes if it were a significant factor. While the plots suggested in this comment would be fascinating to consider in a study with a more appropriate experimental design, here the would only illustrate this unsurprising negative result, and distract from the main purpose of this study.

Changes to the Manuscript: We did not make any changes in response to this comment.

Referee Comment 51: Lines 269-272: Background on the analysis by Moore et al., 2009 should be introduced in the beginning of the manuscript. Also, I had a hard time following the organization of this paragraph. Maybe split paragraph into discussion of cliff-dominated systems $T_r \_0$, and soil-mantled systems $T_r$ Âż $T_p$?

Author Answer: Agreed.

Changes to the Manuscript: Background on the Moore et al. (2009) study has been included in the expanded site description section in the methods. This discussion paragraph has been revised to more clearly distinguish between the two end member cases.

Referee Comment 52: Lines 275-276: Add reference to Attal et al., 2015 and Roda-Boluda et al., 2018

Author Answer: Agreed.

Changes to the Manuscript: These two references are now cited in this section.

Referee Comment (Set) 53: Line 276: a bit unclear what is meant by "sites", yes if the site is only a talus and cliff system, the weathering zone is very thin or only along fractures and $T_r = 0$, but there is a problem if you scale this up to a full catchment (which is relevant for landscape evolution), full catchments typically include a mix of bare-bedrock and soil-mantled hillslopes (see DiBiase et al., 2012 https://doi.org/10.1002/esp.3205 ; Milodowski et al., 2015 www.earth-surf-dynam-discuss.net/3/371/2015/doi:10.5194/esurfd-3-371-2015 ; Neely et al., 2019 https://doi.org/10.1016/j.epsl.2019.06.011 ) Line 276: I'm struggling to understand the spatial scale over which $T_r \_0$. At the catchment scale, catchments are rarely 100% bare bedrock (there's still some soil). At what point does the catchment system switch from $T_r = 0$ to $T_r$ Âż $T_p$? Lines 279 – 311: I think it would be useful to define the spatial scale over which these timescales ($T_r$ and $T_p$) are being assessed. Consider

breaking this up into paragraphs that address this issue at the cliff/hillslope scale, and then the catchment scale (which mixes sediment from cliffs and soil-mantled hillslopes)? At the scale of a single cliff or hillslope, I think this ratio/analysis may be helpful, but at the scale of a steep catchment with cliffs, usually hillslopes are a mix of soilmantled slopes, talus slopes, and bare-bedrock cliffs. It becomes challenging to define 'H' in these settings or ensure that 'E' is the same for soil-mantled slopes and cliffs. (See Neely et al., 2019 https://doi.org/10.1016/j.epsl.2019.06.011 and Neely and DiBiase preprint https://doi.org/10.1002/essoar.10502617.2 and Sklar et al., 2020 https://doi.org/10.1002/esp.4849 ) I do think this is a useful discussion point to link the study to past work, but maybe this should be reframed? - i.e. at the scale of exposed cliffs and downslope talus, $T_r$ Âń $T_p$ and sediment grain size reflects latent clast sizes. The emergence of cliffs in steep landscapes then implies a larger supply of sediment with grain sizes that mirror bedrock fracture spacing?

Author Answer: These are excellent questions and points about the issue of scale in applying this conceptual framework. We would certainly expect spatial variability within a catchment, even along a single hillslope, in the relative influence of latent sediment size and weathering on the size of sediments produced at any given location. Each of the possible controlling factors may vary on a different length scale, creating complex patterns of variation. Alternatively, relatively simple gradients might arise due to gradual covariation of factors such as erosion rate and local climate in a high relief catchment underlain by uniform rock. We propose this framework as a conceptual tool to be applied primarily at the scale of a locally homogenous sediment production regime, however large or small that might be. Hopefully this framework will be useful for understanding and predicting spatial variability in initial sediment size, once the controls within a given sediment production regime are well understood.

Changes to the Manuscript: We have expanded this discussion to include the issues of scale and spatial variability.

Referee Comment (Set) 54: Also, it's not clear to me how long the sediment in the talus

cones remains in the talus cones? Tr could be very long on the talus cones, but lack of biota, water retention, and soil could lower the efficacy of weathering on these cones. So the sediment is detached, sits in the weathering zone as mobile regolith, but there is little fining occur- ring? For example, post-glacial rock piles and boulder fields may not develop soil for tens to hundreds of thousands of years despite being in temperate climates on low slopes (see Denn et al., 2017 https://par.nsf.gov/servlets/purl/10098651) In the boulder field setting, Tr Âż Tp, but the latent particle size still dominates? – maybe this is a special case, but relict boulder fields are common in post-glacial landscapes such as the ones maybe studied by Moore et al., 2009??

Author Answer: Our conceptual framework is only intended to apply to the question of what controls the initial sediment size, and not how sediment size evolves afterward. However, the long-term fate of talus is a very interesting question! Initial size and degree of pre-detachment weathering will certainly influence how rapidly size evolves within a talus deposit, however this topic is beyond the scope of this paper.

Changes to the Manuscript: We did not make any changes in response to this comment.

Referee Comment 55: Line 305: transition to patchy soil cover in these landscapes may be better described by DiBiase et al., 2012 https://doi.org/10.1002/esp.3205 , Heimsath et al., 2012 http://www.nature.com/doifinder/10.1038/ngeo1380 , and Neely et al., 2019 https://doi.org/10.1016/j.epsl.2019.06.011

Author Answer: Agreed.

Changes to the Manuscript: The suggested citations have been added.

Referee Comment 56: Line 305: not sure where these patchy-soil cover landscapes fit in the context of figure 7, because some of the hillslopes within these landscapes are Fig. 7a, and some are Fig. 7b. Also change "DiBiasi" to "DiBiase"

Author Answer: See response to comment set 53 above

Changes to the Manuscript: Misspelling of DiBiase has been fixed throughout the revised manuscript.

Referee Comment 57: Line 306-307: I'm not sure about this point with the difference in fracture detection resolution and fracture spacing measurement technique (scanline vs short-axis between fracture bound block).

Author Answer: Agreed.

Changes to the Manuscript: This passage has been revised as noted in the response to comment 53 above.

Referee Comment 59: Lines 308-311: I think this point has been demonstrated in other analyses: Attal et al., 2015; Roda-Boluda et al., 2018; Neely and DiBiase preprint. The data from this study does not really show much soil weathering (unless some data is replot to show sediment fining in the talus piles due to climate variables??), so maybe this point should be rephrased, cited, or removed?

Author Answer: Agreed.

Changes to the Manuscript: This passage has been revised to place our conceptual framework in the context of the results of these other studies, cited here.

Referee Comment 60: Line 483: why 13000? Is there are source/data suggesting this is when deglaciation occurred near these cliffs? Should this number be the same for each cliff?

Author Answer: This is the regional value used by Moore et al. (2009) in their calculations of cliff retreat rate.

Changes to the Manuscript: In the expanded study site subsection at the beginning of the methods section, we include this age value in the review of the Moore et al. (2009) results, along with a relevant citation.

Referee Comment 61: Figure 1: I think it would be useful to expand the study map

figure into its own figure (new figure 1) and show the geologic context (bedrock lithology and distribution of active and inactive faults). Can you add a scale bar to each panel? I'm concerned about the road in the bottom two panels, and I'm not sure where in panel CB-1 and MT-39 the measurements occurred (highlight the transects a bit?)

Author Answer: We appreciate the suggestion to elaborate on the geologic context, however the study was not designed to explain why fracture spacing varies, only how initial sediment size might be related to fracture spacing. The suggested figure would be more appropriate for a different study, for example one designed to test the influence of fault proximity on fracture spacing. Scale bars would be impractical for most of these site photos, given the considerable foreshortening. These photos are intended to provide general context, to give the reader a feel for the environmental settings, not to document the exact measurement locations. We do that in Figure 2 for one site as an example.

Changes to the Manuscript: We did not make changes in response to this comment.

Referee Comment 62: Figure 2: add a scale bar to bottom right image? Could use this figure to illustrate how the scan-line fracture spacings were measured?

Author Answer: Agreed. A sense of scale would improve the bottom right image. Annotating the bottom left figure would help illustrate how the scan-line fracture spacings were measured.

Changes to the Manuscript: For bottom right image, the caption has been revised to include the length of the tape transect and to point out the person in red jacket for scale. We have revised the bottom left image to include annotations identifying the fracture spacings measured.

Referee Comment 63: Figure 4: add p-values referenced in text that compare to the 1:1 line to panels?

Author Answer: Labeling the regression lines with p values could be misleading. That

might imply that the null hypothesis being tested is that the slope of the line is zero. We have revised the caption instead.

Changes to the Manuscript: Caption revised be more precise about the finding of no-significant difference between the regression line slopes and the 1-to-1 line.

Referee Comment 64: Figure 5: mark resolution detection limits on x-axis of panels?

Author Answer: The detection limit is implicit as no data exist that fall below the detection limits. Adding a vertical line at the equivalent of 2 mm would clutter already busy graphs with information of low importance. In addition, it could be confusing because the x-axis is normalized by the mean.

Changes to the Manuscript: We did not make changes in response to this comment.

Referee Comment 65: Figure 6: I find the C:A notation a bit confusing and prefer C/A, B/A, A-B/A-C notation (just my preference though). Nice figure!

Author Answer: We used the same notation as in the original Sneed and Folk (1958). Glad you liked the figure!

Changes to the Manuscript: We did not make changes in response to this comment.

Referee Comment (Set) 66: Figure 7: I'm a bit confused about the scale/dimensions. Make sure to have (A) (B) and (C) markers on panels? I'm not sure about the assumption that the erosion rate and slope are the same despite different fracture spacing or weathering zone thickness (in caption and in discussion section). For example, soil transport rate for a given slope may change in response to different clast size (larger blocks require steeper slopes to move at same erosion rate – see Neely et al., 2019 https://doi.org/10.1016/j.epsl.2019.06.011 , DiBiase et al., 2018 https://doi.org/10.1016/j.epsl.2018.10.005 ) If climate changes drive difference in weathering zone thickness, that may also change the hillslope erosion rate – see Owen et al., 2011 https://doi.org/10.1002/esp.2083 ) I'm not sure the assumptions in this conceptual model are realistic because weathering zone thickness is often coupled

to hillslope erosion rate, fracture spacing, and climatic/ecologic factors?

Author Answer: These are simple cartoons intended to illustrate the concepts. They are not intended to be scaled diagrams or to depict actual landscapes. Our intention is to focus here on just two of the many factors that can vary to create large differences in the time-scale ratio: fracture spacing in unweathered bedrock and the climate-controlled thickness of the weathering zone. According to the definitions of the two time scales, erosion rate is an important factor, however it cancels out of the ratio. This does not mean erosion rate doesn't matter, only that the influences of particle size and weathering zone thickness both scale relative to erosion rate. Other cartoons could be created to illustrate the conceptual framework that might vary erosion rate and slope, however we prefer to use this version to focus on particle size and weathering zone thickness.

Changes to the Manuscript: We have added letter labels to each panel.

Referee Comment 67: — These are a lot of edits! But I think most of these items can be addressed or clarified? The dataset is a valuable foundation, and I think with some more context and specific language surrounding the methods, this will be a very useful study that can be pulled into a lot of future research! Hopefully this isn't discouraging or too intimidating.

Author Answer: We are enormously grateful to the Referee for these valuable and voluminous comments! We have done our best to make revisions to satisfy all those comments that could reasonably be addressed. Many of the remaining comments and suggestions are insightful and creative, but are outside the scope of this study.

---

## Author Response (AR2)

Author response to review comments from Associate Editor on manuscript:
 "Sediment size on talus slopes correlates with fracture spacing on bedrock cliffs: Implications for predicting initial sediment size distributions on hillslopes" by Joseph P. Verdian et al.

June 25, 2021

We are grateful for these astute and targeted comments, which highlighted a number of important shortcomings in the previous draft. We agree with all of the points raised, and have made changes to the manuscript to address every suggested edit and requested clarification. Following these minor revisions, the manuscript is clearer, and we hope, more persuasive. Once again, thank you for these very helpful comments

Review comment 1
Line 14: It isn't really sediment before it detaches. Consider using "blocks" instead.

Author response: Agreed, good suggestion.

Changes to the manuscript: Text revised as suggested.

Review comment 2
Line 22: I would clarify to say this is talus at your sites. I can imagine a postglacial landscape where the talus generation has stopped (or slowed a lot) and there is weathering of the talus field. I don't think you want to give readers the impression this is universally true

Author response: Agreed, talus certainly does weather in other settings.

Changes to the manuscript: Revised text now reads: "In addition, talus at our sites has not undergone much weathering…"

Review comment 3
Line 82-86: In Inyo Creek the coarse material reflects the fracture spacing. What about the fines? And in the San Gabriels and San Jacinto is it only the coarse fraction or the entire distribution that reflects the bedrock spacing? An edit isn't really needed here but if the answers to the above questions are interest please add.

Author response: The production of fine sediments is certainly of interest, and is addressed in the discussion (last paragraph of section 4.1). Adding this topic to the introduction would be a bit of a digression.

Changes to the manuscript: No changes were made in response to this comment.

Review comment 4
Line 93: There isn't frost cracking at the Inyo site? If the California sites are too warm for frost cracking I would say so above.

Author response: Good point. Conditions favorable to frost cracking do occur at the Inyo Creek site. Frost cracking may be implicated in the observed downvalley fining of the finer mode of the hillslope sediment size distribution there.

Changes to the manuscript: We have added a sentence noting the potential role of frost cracking at Inyo Creek: "Variation in the frequency and intensity of frost cracking with elevation at the Inyo Creek site may also be implicated in the downvalley fining of the fine mode of the hillslope sediment size distribution (Riebe et al., 2015; Sklar et al., 2020)."

Review comment 5
Line 148: This seems too strong: temperature can span much more than 10C, the variation in temperature has a large range as well, you could have 15 sites just spanning metamorphic grade, and precipitation, globally, has a much wider range than in the study sites. I do think you've done a good job in selecting a wide range of sites. But I'm not sure if I would expect these to span "much" (I'm not sure what that even means) of the range in sizes. This would be less distracting if you just said "...expected to produce talus spanning a wide range of sizes".

Author response: Agreed.

Changes to the manuscript: Text revised as suggested.

Review comment 6
Line 162: I feel like it would be helpful to readers if you had an image of the scan line and fractures here (like in figure 2).

Author response: Agreed.

Changes to the manuscript: We have added a second call to Figure 2, near the end of this paragraph, specifically directing readers to panel b, which shows a close-up image of a scan line crossing fractures. The revised text now reads: "This yields a distribution of fracture spacings measured as the distance between successive fractures (Fig. 2b)."

Review comment 7
Line 204: Parallel gives the impression you are comparing straight lines (at least to me). Consider changing this to say "the distributions have the same shape but are offset..."

Author response: Curved lines can be considered "parallel" if the distance between lines (measured perpendicular to the lines) remains approximately constant (for example, here's a link to the relevant Wikipedia page https://en.wikipedia.org/wiki/Parallel_curve). We prefer to use

this term, rather than say they have the same shape, because it is more precise. The "same shape" could be interpreted as meaning they are described by the same function. For example, if the "shape" is a cumulative normal distribution, differing standard deviations would produce non-parallel curves even though the shape is the same.

Changes to the manuscript: To clarify what we mean by "parallel" we have added a parenthetical definition "(i.e. offset by a constant distance)".

Review comment 8
Line 215: Awkward phrasing. Are you saying Messenzehl et al didn't test for correspondence? Clarify this sentence.

Author response: Agreed. This mention of Messenzehl et al., 2018 is not accurate, not needed and not appropriate.

Changes to the manuscript: The clause referring to Messenzehl et al., 2018 has been removed.

Review comment 9
Line 229: Interesting. I've not got to the discussion yet but I'm hoping to hear your opinion of whether this is by chance or there is some mechanistic explanation. - Well, now on page 11 I see there is some reference to this but no speculation if there is a physical process responsible. If only a subset of the fractures end up failing when blocks are released could there be some stress release mechanism that explains this offset? I'm not requesting a change to the text but if you did have an opinion it would be interesting to hear it.

Author response: This is indeed an interesting phenomenon! It is beyond the scope of this paper to address the fracture mechanics that must underlie the differential propagation of fractures of differing sizes. However, a simple (mostly geometric) argument can be made that where longer, more persistent fractures extend more rapidly than smaller fractures, they may be more likely to intersect and detach relatively larger blocks containing smaller fractures.

Changes to the manuscript: We have added a sentence to the discussion paragraph addressing this point. "Incomplete exploitation of fractures could occur where longer, more persistent fractures extend more rapidly than shorter fractures, intersecting to detach relatively large blocks"

Review comment 10
Line 294: I don't really understand what you mean here. The data shows the talus is coarser than suggested by the fracture spacing. Please clarify.

Author response: Agreed. This passage needs clarification. This statement is intended to summarize the fundamental observation that particle detachment from these cliffs is dominated mechanisms that exploit the fractures visible on the cliff face, exploiting most if not all fractures.

Changes to the manuscript: The opening sentence to the discussion now reads "The close correlations between talus size and fracture spacing distributions at our sites (Figs. 3–5) suggest that particles are detached by mechanisms that exploit most if not all of fractures exposed on the cliff faces and do not undergo much size reduction due to physical or chemical weathering in talus deposits."

Review comment 11
Line 297: But this contrasts with the Messenzehl et al results, doesn't it? Does frost cracking (caused by segregation ice) control particle size or not?

Author response: Excellent question! Frost cracking undoubtedly occurs at some of our sites, but other mechanisms may be equally or more important. An assessment of the role of segregation ice growth in detaching particles is beyond the scope of this paper. The mention of segregation ice and sub-critical crack growth is not needed, and not helpful.

Changes to the manuscript: The clause mentioning segregation ice growth has been deleted. The sentence now reads "This finding, while limited to our sites, is robust across a wide range of lithologies and weathering conditions, suggesting that it spans a range of processes that could lead to particle detachment and subsequent weathering in talus deposits."

Review comment 12
Line 313: The surface is coarser. How is it possible the two medians would be the same? Clarify please.

Author comment: Agreed, this passage needs clarification. The method we used (taken from Bunte and Abt, 2001) tests whether the finer sediments should be considered a population distinct from the coarser sediments or whether they represent the fine tail of the distribution of a single population dominated by the coarser sizes. The null hypothesis is that there is only one population. To reject the null hypothesis, the fines must be sufficiently incompatible with the tail of the coarse distribution. In our case, they were not.

Changes to the manuscript: This passage has been revised for clarity, and now reads: "…the size distributions of the bulk samples overlap with the fine tail of the talus distribution enough that they could be combined using established techniques (Bunte and Abt, 2001) into a single continuous distribution. In this case, fine particles are sufficiently numerous on the talus surface that the statistics of the measured surface size distribution would not be affected by loss of a fraction of the fine-tail particles to the interstitial pores."

Review comment 13
Line 320: Say if it reduces the offset.

Author comment: Agreed, this is a vague way of reporting the outcome of this calculation, which showed no significant difference in the offset between the two methods.

Changes to the manuscript: The revised sentence now reads "However, there is no significant difference in the offset when we apply this approach to our data."

Review comment 14
Line 328: Is the conversely needed here if you already said "however"?

Author comment: Agreed.

Changes to the manuscript: The "conversely" has been removed.

Review comment 15
Line 355-358: This could be clearer. If I understand the argument, the talus slopes have no "regolith" (defining regolith in this context is a little challenging) whereas the California sites from other studies do. I guess you mean it is not all loose blocks in these other studies and there are patches of finer material that hold water and promote weathering. Where do these patches come from? I think this part needs a few more sentences of explanation.

Author comment: Agreed, this passage needs a clearer explanation of the comparison between the cliff-talus setting and the bedrock slopes studied in the other California sites.

Changes to the manuscript: Several sentences have been added to provide a clearer explanation. The revised text now reads "However, variations in particle size across these sites can also be explained by differences in weathering that are driven by variations in the fractional coverage of regolith. In these steep mountain landscapes, the rough surfaces of bedrock hillslopes provide locations favorable to transient storage of coarse particles produced on adjacent slopes. During storage, particles are subject to physical weathering processes, such as frost cracking and thermal stresses, and chemical weathering processes aided by the presence of water and vegetation. In contrast, the relatively smooth and nearly vertical cliff faces at our study sites lack locations favorable to transient particle storage, and the adjacent talus surfaces are well-drained and minimally vegetated. Hence, we conclude that the offset toward finer sizes evident at the Inyo Creek, San Gabriel, and San Jacinto sites is due in part to substantial weathering not experienced on the bare cliff faces and talus slopes at our sites.

Review comment 16
Line 376: This needs an "e.g." since this is only a subset of papers reporting this.

Author response: Agreed.

Changes to the manuscript: "e.g." has been added as suggested.

Review comment 17
Line 390: I don't think "all of the sediment" is the same as "one layer". Surely the talus slope is more than one layer thick.

Author comment: This sentence needs clarification. The inference is that at the site with the most slowly retreating cliff face, the volume of talus that has accumulated over the past 13,000 years was produced by a cliff-normal, average depth of erosion of the cliff face equal to the median spacing between fractures. The talus pile is presumably many particles thick, except at the toe, but must occupy an area smaller than the area of the cliff face area that eroded to produce the particles. In other words, the talus slope at this site is not one layer thick, but was produced by one-layer's worth of particle detachment from the bedrock cliff.

Changes to the manuscript: We have made several edits to clarify this point. The revised passage now reads "Application of Equation 1 to data from our sites indicates that $T_P$ is as short as 88 years for detachment of one layer of 60-mm diameter latent particles from the cliff at EP-26, the most rapidly eroding cliff face. At the most slowly eroding cliff face, CB-1, Equation 1 suggests that it took 16,500 years to detach one layer of 330-mm diameter particles, indicating that the entire post-glacial accumulation time and more was needed to detach a single layer of latent particles with the characteristic median size (Table 2). The calculated $T_P$ at the remaining talus-cliff pairs in the Sierra Nevada sites is less than 13,000 years, consistent with the assumption in the cliff retreat rate calculations of Moore et al. (2009) that all of the sediment now contained within the talus piles was produced after the glaciers retreated.

Review comment 18
Line 393: Why would the cliff retreat rate (used to calculate T_P be the same as the surface erosion rate? I think equation (2) needs an erosion ratio between cliff retreat and vertical erosion on the talus/soil

Author response: This is a very helpful comment because it shows that in the previous draft we failed to adequately explain the definitions of the two terms involved in the derivation of equation 2.

The definition of T_P is independent of the orientation of the eroding surface, and is based on the simplifying assumptions erosion occurs normal to the bedrock surface and that the median fracture spacing characterizes the depth of erosion that results from detachment of a typical particle. The definition of T_R is also independent of the orientation of the eroding surface, and is based on the simplifying assumption that rock mass is exhumed on a vector normal to the eroding surface, passing through a weathering zone with a thickness defined along a line also oriented normal to the eroding surface. In applying this framework to our eroding cliff sites, cliff retreat rate is the relevant erosion rate for both particle production and bedrock weathering zone thickness. This is the rate at which rock mass is exhumed toward the eroding surface where particles are produced. Much like a symmetrical weathering rind on the surface of a large clast,

weathering of the rock mass the near the eroding cliff face can reasonably be assumed to occur over some thickness normal to the free face.

Erosion of the talus slope is not relevant to calculating the time-scale of particle production nor is it relevant to calculating the time-scale of pre-detachment weathering within the bedrock, because particles do not become part of the talus slope until after they are detached from the rock mass.

Changes to the manuscript: We have made numerous changes to sections 4.2 and 4.3 to clarify these points. The changes include: Adding an equation (the new Equation 1) for residence time, with accompanying text to explain; replacing the term regolith with more precise language; additional text to clarify that erosion rate ($E$) applies to the bedrock cliff face and particle production surface and not to the depositional talus surface; additional text to more clearly explain the conceptual framework of pre-detachment weathering occurring during exhumation of rock normal to the surface where particles are produced.